SPECIAL ISSUE
LIFELONG DEVELOPMENT

# Nr4a1 modulates inflammation and heart regeneration in zebrafish

Dong Feng[1,2,*,‡], Yanhan Dong[1,2,‡], Yiran Song[1,2], Nicholas Yapundich[1,2], Yifang Xie[1,2], Brian Spurlock[1,2], Tingting Lyu[1,2], Landry Kuehn[1,2], Li Qian[1,2] and Jiandong Liu[1,2,§]

## ABSTRACT

Recent findings have highlighted the complex role of inflammation in zebrafish heart regeneration, demonstrating that although inflammation is essential for initiating transient fibrosis and tissue repair, chronic inflammation, and unresolved fibrosis, could impede full regenerative recovery. In this study, we identified the nuclear receptor Nr4a1 as a crucial regulator of this regenerative process in zebrafish. Loss of Nr4a1 function led to a prolonged and excessive inflammatory response, disrupted neutrophil migration, delayed fibrin clearance, and ultimately impaired heart regeneration. Transcriptome-wide RNA-seq analysis at different injury stages revealed molecular disruptions associated with dysregulated inflammation and fibrosis in *nr4a1* mutants. Notably, partial inhibition of the pro-inflammatory cytokine Tnfα rescued heart regeneration in the *nr4a1* mutants, highlighting the therapeutic potential of modulating inflammation. Our findings suggest that Nr4a1 plays a crucial role in orchestrating the immune response during heart regeneration and may serve as a valuable target for enhancing cardiac repair following injury.

KEY WORDS: Cardiac regeneration, Nr4a1, Macrophage, Neutrophil, Zebrafish

## INTRODUCTION

Heart diseases, particularly myocardial infarction (MI), remain a leading cause of death worldwide due to the limited regenerative capacity of the adult mammalian heart (Lodrini and Goumans, 2021). After MI, the mammalian heart permanently loses millions of cardiomyocytes (CMs), triggering a complex inflammatory response, leading to scar formation and eventual heart failure (Carrillo-Salinas et al., 2019; Litviňuková et al., 2020). In contrast, adult zebrafish exhibit exceptionally robust regenerative responses to cardiac injury through the activation of CM proliferation, which effectively replaces lost or damaged myocardial tissue (Poss et al., 2002; Sanz-Morejón and Mercader, 2020). This robust regenerative response highlights the zebrafish as an invaluable model for investigating the underlying principles of heart regeneration that

could inform therapeutic interventions (Laflamme and Murry, 2011; Tzahor and Poss, 2017).

The immune response is essential for initiating wound healing and shaping tissue outcomes, leading to either a regenerative or pro-fibrotic phenotype (Wynn, 2008; Neher et al., 2011; Cooke, 2019). Following cardiac injury, innate immune cells, including neutrophils and macrophages, are initially activated and infiltrate the injured area (Carrillo-Salinas et al., 2019). Neutrophils are among the first responders to injury, which are crucial for the initial inflammatory phase (Kolaczkowska and Kubes, 2013; Nauseef and Borregaard, 2014). They help to clear tissue debris and release signaling molecules that recruit additional immune cells to the site of injury (Kolaczkowska and Kubes, 2013; Peiseler and Kubes, 2019). However, their prolonged presence can be detrimental, as they release proteases and reactive oxygen species that can cause further tissue damage (Greenlee-Wacker, 2016; Herrero-Cervera et al., 2022).

Macrophages, which are activated after neutrophils, play a crucial role in clearing debris, propagating inflammation and remodeling the extracellular matrix (ECM) in a timely and spatially coordinated manner (Braga et al., 2015; Watanabe et al., 2019; Witherel et al., 2019). They not only clear apoptotic cells and matrix debris but also secrete cytokines and growth factors that modulate inflammation and promote tissue repair (Watanabe et al., 2019; Westman et al., 2020). The timing and regulation of macrophage recruitment are crucial for effective healing. Delayed macrophage recruitment can result in neutrophil retention, compromised CM proliferation and impaired scar resolution (Silva, 2011; Braga et al., 2015; Ma et al., 2021; Yang et al., 2023). Conversely, excessive or prolonged inflammatory responses can lead to pathological cardiac remodeling and failed heart regeneration (Schett and Neurath, 2018). During the later regenerative phase after injury, macrophages acquire anti-inflammatory properties, marked by the secretion of anti-inflammatory signals and matrix metalloproteases (MMPs), essential for resolving inflammation and regressing fibrosis (Cao et al., 2014; Braga et al., 2015; Witherel et al., 2019). However, the factors driving the balanced immune response and the fine-tuned interplay among multiple cell types remain largely unexplored.

Nr4a1 (also known as Nur77) is a nuclear receptor responsive to external stimuli (Chen et al., 2020) primarily expressed in myeloid cells and closely associated with the functions of macrophages and neutrophils (Hanna et al., 2011; Tacke et al., 2015; Prince et al., 2017). In mice, Nr4a1 mutation has been shown to cause excessive inflammation and exacerbated atherosclerosis (Hanna et al., 2012; Ipseiz et al., 2014). Additionally, loss of Nr4a1 function retains the activation and immigration of neutrophils in a brain ischemic stroke model (Strecker et al., 2022 preprint). Although these studies indicate Nr4a1 as a potent modulator of inflammation, its role in the immune response following cardiac injury in zebrafish remains unknown.

[1]Department of Pathology and Laboratory Medicine, University of North Carolina, Chapel Hill, NC 27599, USA. [2]McAllister Heart Institute, University of North Carolina, Chapel Hill, NC 27599, USA.
*Present address: Department of Internal Medicine, Yale University, New Haven, CT 06510, USA.
‡These authors contributed equally to this work

§Author for correspondence ( jiandong_liu@med.unc.edu)

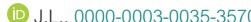 J.L., 0000-0003-0035-3570

**DEVELOPMENT**

In this study, we demonstrate that depletion of *nr4a1* leads to a prolonged inflammatory response, impaired neutrophil migration, delayed fibrin clearance and, ultimately, failed heart regeneration. RNA-seq analysis at multiple injury stages reveals distinct molecular changes associated with the disrupted inflammatory and fibrotic processes in the mutants. Notably, the compromised heart regeneration in *nr4a1* mutants can be partially rescued by manipulating the cytokine tumor necrosis factor (Tnfα) to inhibit excessive inflammation. Our findings indicate that the nuclear receptor Nr4a1 may contribute to heart regeneration by modulating the inflammatory response, suggesting that Nr4a1 is a potential target for therapeutic strategies to improve cardiac repair.

## RESULTS

### Dynamic expression of Nr4a1 during zebrafish heart regeneration

We first examined the temporal expression pattern of *nr4a1* in the cryoinjury model of zebrafish heart regeneration (González-Rosa and Mercader, 2012). Notably, compared to its expression in uninjured hearts, *nr4a1* exhibited significant upregulation at 2 days post-cryoinjury (dpci), followed by decreased expression at 7 dpci and a resurgence in expression at 14 dpci (Fig. 1A). Subsequently, its expression decreased again at 21 dpci (Fig. 1A). To delineate the cell types where Nr4a1 is expressed during cardiac regeneration, we sought to generate a stable knock-in (KI) transgenic line where the

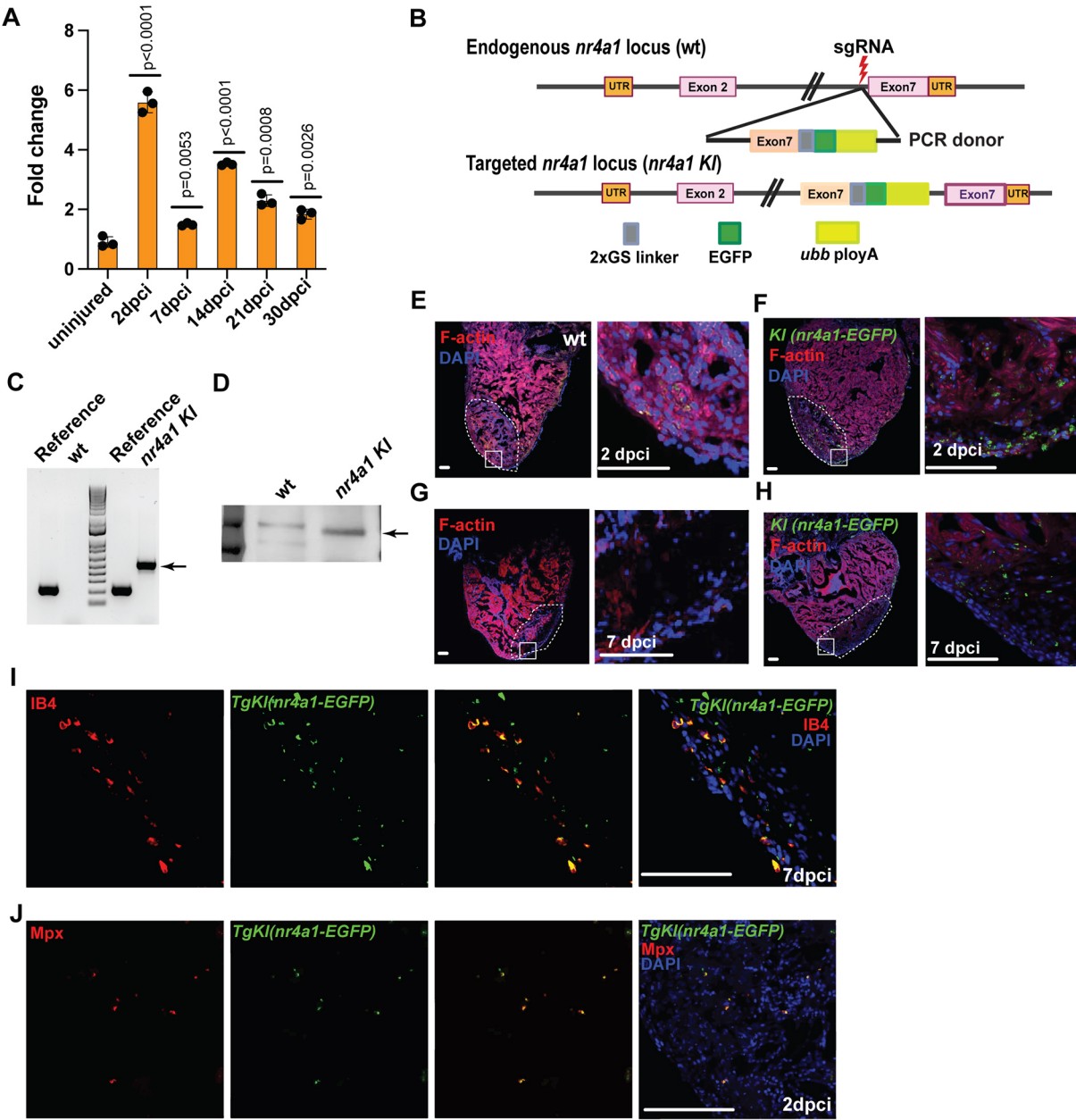

**Fig. 1. Dynamic expression of Nr4a1 after cardiac injury.** (A) Expression pattern of *nr4a1* at different stages post-injury. The fold change is calculated relative to the expression in uninjured wild-type (wt) hearts. (B) The schematic for generating *TgKI(nr4a1:eGFP)*. (C) Detection of *nr4a1:eGFP* fusion mRNA. The 518 bp band indicates correct *nr4a1:eGFP* fusion cDNA. (D) Western blot analysis of *TgKI(nr4a1:eGFP)* to identify the fusion Nr4a1:eGPF protein (88kD). (E-H) Expression of Nr4a1, as revealed by GFP antibody staining, in 2 and 7 dpci hearts. (I) IB4 staining to localize Nr4a1 expression in macrophages within 7 dpci hearts. (J) Anti-Mpx antibody staining to localize Nr4a1 expression in neutrophils within 2 dpci hearts. *P*-values<0.05 were considered statistically significant (two-tailed unpaired *t*-test). Scale bars: 60 μm.

endogenous Nr4a1 protein is tagged with eGFP at its C-terminal end. To this end, we applied the recently published knocking-in tagging method and designed a guide RNA (gRNA) to target the last intron of *nr4a1* (Levic et al., 2021). Co-injection of the Cas9 protein, the gRNA and the repair donor enabled the integration of the donor DNA fragment containing the eGFP sequence into the *nr4a1* locus through non-homologous end joining (Fig. 1B). We obtained a stable *TgKI(nr4a1-eGFP)* knock-in line in which Nr4a1-eGFP is expressed properly, as confirmed by RT-PCR with primers for the fusion gene and by western blot analysis using GFP antibody (Fig. 1C,D). Using GFP as a proxy for Nr4a1 expression, we performed immunohistochemistry on cardiac sections of uninjured and injured hearts. We found that, in contrast to the undetectable level of GFP expression in the uninjured hearts, Nr4a1 expression was observable around the injured area at 2 and 7 dpci (Fig. 1E-H). We then examined GFP expressions in the *TgKI(nr4a1-eGFP)* hearts by colocalizing it with markers of macrophages (IB4) and neutrophils (Mpx). GFP expression was readily observable in both cell types (Fig. 1I,J) (Lai et al., 2017). To further investigate the dynamics of Nr4a1 relative to that of macrophages and neutrophils, we performed triple immunostaining for GFP, IB4 and Mpx at various time points post-injury. As shown in Fig. S1A-C, the Nr4a1 expression peaked at 2 dpci and then gradually decreased by 7 dpci, partially mirroring the expansion dynamics of IB4$^+$ macrophages and Mpx$^+$ neutrophils. Subsequent quantification of Nr4a1$^+$IB4$^+$ or Nr4a1$^+$Mpx$^+$ cells further confirmed the expression of Nr4a1 in both cell types (Fig. S1D,E). Interestingly, we also observed a population of triple-positive cells (Nr4a1$^+$IB4$^+$Mpx$^+$), which may reflect macrophage-mediated phagocytosis of neutrophils during heart regeneration (Harrison et al., 2019). In summary, these findings indicate that *Nr4a1* is dynamically expressed in multiple innate immune cell types.

### Loss of *nr4a1* function impaired zebrafish heart regeneration

To elucidate the potential role of Nr4a1 in heart regeneration, we generated *nr4a1* mutant using the CRISPR/Cas9 system. We generated a mutant allele containing a one base pair insertion (*nc40*) in the second exon of the *nr4a1* gene (Fig. 2A). This mutation results in a premature stop codon, predicted to truncate the protein upstream of the DNA- and ligand-binding domains (Fig. 2A). Nevertheless, *nr4a1* mutant fish survived to adulthood without overt developmental and growth defects. We then performed cryoinjury and evaluated cardiac regeneration of the injured control and mutant hearts using acid fuchsin orange-G (AFOG) staining (Fig. 2B). Remarkably, the mutant fish displayed compromised heart regeneration with a significant increase in the wounded area compared to the control fish (Fig. 2D), suggesting that loss of *nr4a1* function adversely affected the ability of the heart to repair the damaged myocardium. Furthermore, PCNA labeling revealed that the mutant hearts exhibited a significantly reduced number of Nkx2.5$^+$ proliferating CMs compared to control hearts at 7 dpi (Fig. 2C,E) (de Sena-Tomás et al., 2022; Dong et al., 2024). At 30 and 60 dpci, the *nr4a1* mutants exhibited more significant gaps in the myocardial wall, indicative of impaired regeneration (Fig. 2F-I). These observations indicate that CM proliferation after cardiac injury is impaired in mutant hearts.

### Neutrophil migration and activity were disrupted in the *nr4a1* mutant

Given the prominent expression of Nr4a1 in neutrophils, we used the *Tg(lyz:eGFP)* transgenic line to monitor the infiltration and resolution of neutrophils following cardiac injury. In the injured control hearts, a substantial accumulation of neutrophils at the wounded area was evident at 6 h post-cryoinjury (hpci) (Fig. 3A,B,G). At 1 dpci, we observed a decrease in the number of neutrophils (Fig. 3C,G) at the injury area, followed by a slight increase at 2 dpci (Fig. 3D,G). As the inflammation subsided, a significant reduction in neutrophil number was observed at 7 and 14 dpci (Fig. 3E-G). Next, we examined neutrophil infiltration and resolution in the *nr4a1* mutant hearts (Fig. 3A-D). In contrast to the control hearts, we observed fewer neutrophils at the injury site of the *nr4a1* mutant hearts at 6 hpci and 2 dpci (Fig. 3H,I), suggesting that neutrophil accumulation at the injury area was compromised in the absence of *nr4a1* (Fig. 3J). Despite the reduced number of neutrophils in the early stage, a significantly higher number of neutrophils persisted around the wounded area in the mutant heart during the later stages (Fig. 3E-G). This finding was further confirmed using *Tg(lyz:eGFP)* transgenic line and Mpx antibody staining on the heart sections (Figs S2 and S3). These results indicate that *nr4a1* expression is crucial for both neutrophil infiltration and clearance, impairments that may lead to sustained tissue damage.

### The *nr4a1* mutant exhibited an increased and widespread accumulation of inflammatory macrophages

We next evaluated the effect of *nr4a1* loss on macrophage behavior following cardiac injury. Using *TgBAC(tnfa:eGFP)* transgenic line, alone or in combination with IB4 staining, we examined the distribution of inflammatory macrophages in both control and mutant hearts at different stages. A substantial accumulation of inflammatory macrophages was observed at the injury area of the control hearts at 7 dpci (Fig. 4A,B; Fig. S4A,B). Notably, the *nr4a1* mutant exhibited a higher accumulation of inflammatory macrophages in the wounded area compared to the control (Fig. 4B,D; Fig. S4A,B). At 14dpci, while the overall macrophage number decreased in the control heart, we observed a persistent presence of inflammatory macrophages in the wound area of the *nr4a1* mutant (Fig. 4C,E). A similar observation was obtained on cardiac sections (Fig. S4C-E). Consistent with increased and persistent accumulation of proinflammatory macrophages in the mutant hearts, we found significantly higher expression levels of the inflammatory-related genes *tnfa and acod1* across all post-cardiac injury stages in the *nr4a1* mutant heart than in the control hearts (Fig. 4F-H). Concurrent RNAscope *in situ* hybridization for *tnfa* and immunostaining for GFP on 7 dpci cardiac sections, followed by quantification of *tnfa* mRNA signals normalized to *tnfa:eGFP*$^+$ cells, further revealed a significant increase in *tnfa* mRNA levels at the injury area in *nr4a1* mutant hearts when compared to controls (Fig. S5). Additionally, using the *Tg(mpeg1:eGFP)* transgenic line that labels all macrophages, we also observed an increased number of macrophages in the mutant hearts at 7 and 14 dpci (Figs S6 and S7). Taken together, these findings strongly suggest that *nr4a1* deficiency leads to a heightened inflammatory response following injury, which may contribute to the compromised cardiac regeneration observed in the mutant hearts (Fig. 4I).

### The *nr4a1* mutant displayed chronic and unresolved fibrosis

In adult zebrafish, cardiac regeneration involves a transient phase of fibrosis that gradually resolves as regeneration progresses (Sánchez-Iranzo et al., 2018; Ryan et al., 2020; Sanz-Morejón and Mercader, 2020). The extracellular proteins in the fibrotic tissue, such as collagens and periostin, are secreted by fibroblasts derived from *Tg(tcf21:nucGFP)*$^+$ cells localized at the wound site. We thus examined the distribution of *Tg(tcf21:nucGFP)*$^+$ cells between

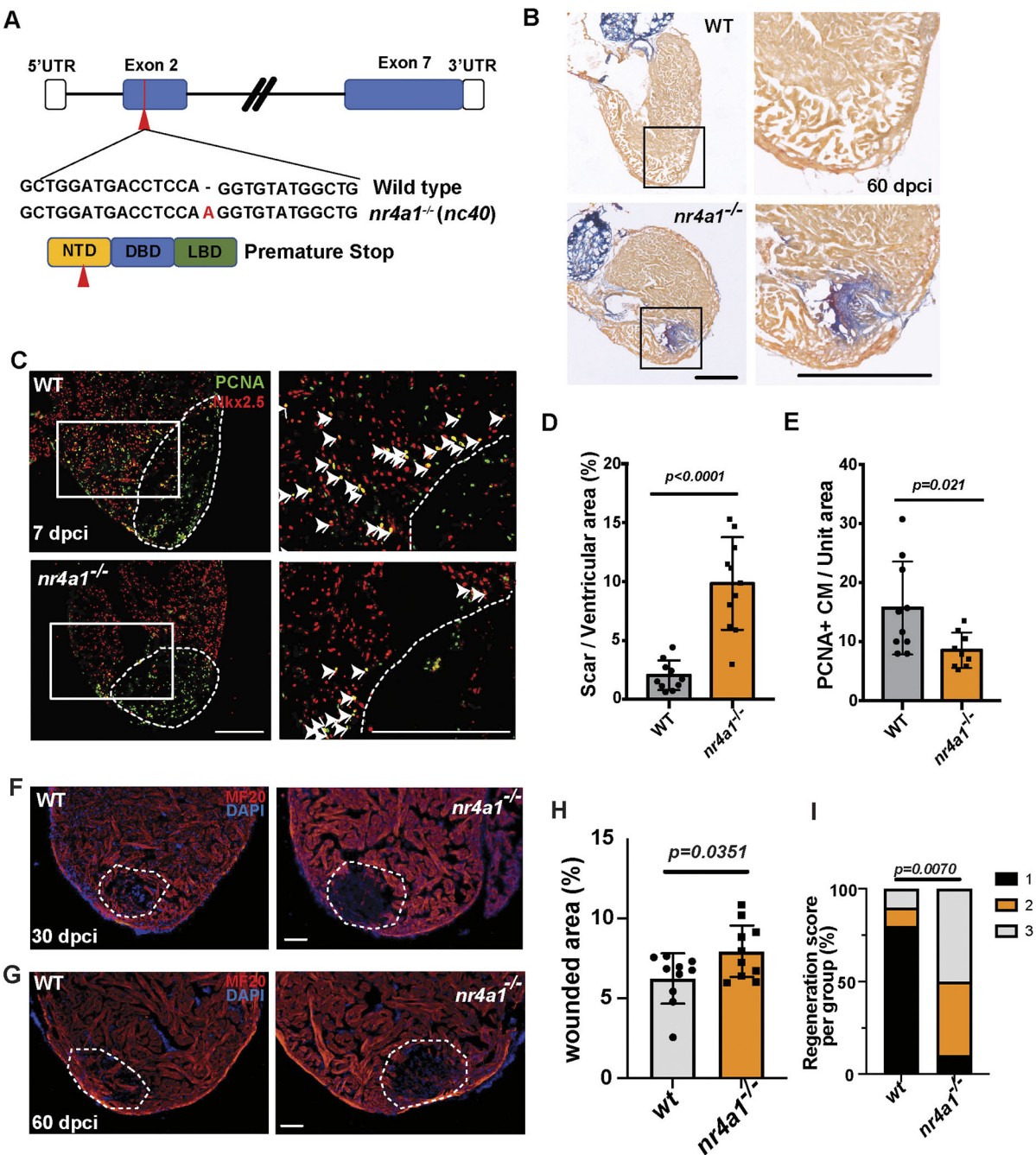

**Fig. 2. Impaired cardiac regeneration in *nr4a1* mutants.** (A) Diagram depicting the generation of *nr4a1* mutants, illustrating the mutations at both the genome and protein levels. (B) AFOG staining performed on wild-type (WT) and *nr4a1* mutant samples at 60 dpci. Boxed regions show the position of magnified images on the right. (C) PCNA/Nkx2.5 double immunostaining to detect proliferating cardiomyocytes at 7 dpci on WT and *nr4a1* mutant cardiac sections. Arrowheads indicate the proliferating cardiomyocytes. Boxed regions show the approximate positions for quantification. (D) Quantification of scar area based on AFOG staining. (E) Quantification of colocalization between PCNA and Nkx2.5 per unit area for both WT and *nr4a1* mutant samples. (F,H) Section images of WT or *nr4a1* mutant ventricles at 30 dpci assessed for muscle recovery (F) and quantification of injured area (H). (G,I) Myocardial regeneration was measured at 60 dpci in WT and *nr4a1* mutants, respectively. Ten individual samples in each group were examined. Myocardial regeneration is categorized as follows: 1, complete regeneration of a new myocardial wall; 2, partial regeneration; and 3, a strong block in regeneration. Dashed lines (C,F,G) show the wounded area. *P*-values<0.05 were considered statistically significant (two-tailed unpaired *t*-test, D,E,H; Chi-squared test, I). Points in D, E and H show the individual samples. Data are mean±s.e.m. Scale bars: 275 μm (B); 250 μm (C); 100 μm (F,G).

control and mutant hearts after cardiac injury. The uninjured and injured control hearts showed a similar distribution of *Tg(tcf21: nucGFP)*+ cells at 7 dpci (Fig. 5A) apart from minor accumulation of *Tg(tcf21:nucGFP)*+ cells at the wounded area in injured control hearts (Fig. S8A). In contrast, a substantial increase in *Tg(tcf21: nucGFP)*+ cells was observed around the wound area in *nr4a1*

mutant hearts by 7 dpci (Fig. S8A,B). At 14 dpci, both control and mutant hearts exhibited substantial accumulation of *Tg(tcf21: nucGFP)*+ cells in the wounded regions (Fig. S8C,D). By 21 and 30 dpci, however, the mutant hearts displayed a significantly higher accumulation of *Tg(tcf21:nucGFP)*+ cells in the wounded regions compared to control hearts, which may correlate with

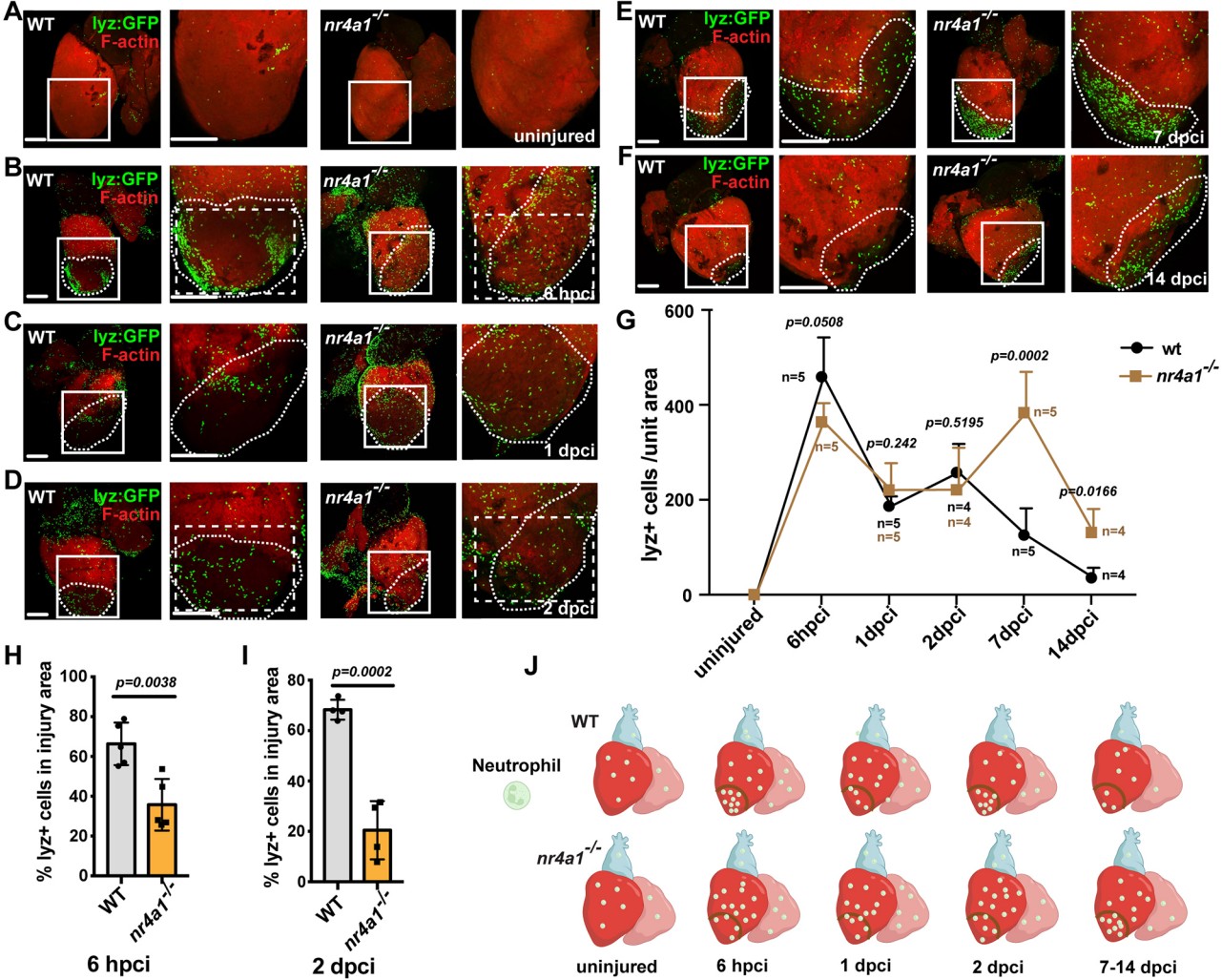

Fig. 3. Defective neutrophil migration in *nr4a1* mutant. (A-F) Left: distribution of neutrophils in uninjured wild-type (WT) hearts and WT hearts at 6 hpci and 1, 2,7, and 14 dpci. Right: distribution of neutrophils in uninjured *nr4a1* mutant hearts and *nr4a1* mutant hearts at 6 hpci and 1, 2,7, and 14 dpci. Boxed regions show the approximate positions for cell number quantification, and position of magnified images. Dashed box regions show the approximate positions of neutrophils around the wounded area. (G) Temporal dynamics of neutrophil number across multiple time points. Individual sample numbers (*n*) are indicated on the plot. (H,I) Percentage of neutrophil numbers in the wounded area per unit square area. (J) Schematic depicting the temporal dynamics of neutrophil number during heart regeneration. Created in BioRender by Feng, D., 2025. https://BioRender.com/vyhyvzk. This figure was sublicensed under CC-BY 4.0 terms. *P*-values<0.05 were considered statistically significant (two-tailed unpaired *t*-test, G). Data are mean±s.e.m. Scale bars: 275 μm.

compromised myocardial regeneration at the late stage due to the impaired clearance of fibroblasts (Fig. 5B-E). Immunostaining using an anti-Collagen 1 antibody revealed persistent collagen deposition in the mutant hearts up to 30 dpci, the final time point examined (Fig. 5F-I; Fig. S9A-D). In line with the formation of a permanent scar at 60 dpci, *nr4a1* mutants displayed substantial accumulated α-SMA⁺ myofibroblasts (Fig. 5J-M). To gain molecular insights into the dysregulated fibrotic program in *nr4a1* mutants, we then performed quantitative PCR analysis for epicardial cell marker *tcf21*, pro-regenerative ECM genes *postnb* and *fn1a*, and pro-fibrotic genes *egr1* and *loxa* (Allanki et al., 2021). Surprisingly, both pro-regenerative and pro-fibrotic ECM genes showed marked and sustained upregulation in the mutant hearts compared to the control (Fig. S10). To corroborate these findings, *in situ* hybridization for *postnb* at 21 dpci showed significantly elevated signals in mutant hearts compared to control (Fig. S9E,F). Collectively, our data indicated that *nr4a1* mutant hearts exhibit increased fibrotic accumulation, which is associated with compromised myocardial regeneration (Fig. 5N).

## Transcriptome analysis revealed molecular changes associated with elevated inflammation, fibrosis and apoptosis in the *nr4a1* mutant

To elucidate the molecular basis of the phenotypes observed in the *nr4a1* mutant, we performed transcriptome-wide bulk RNA-seq on the ventricles from both control and mutant hearts at several key time points (Fig. S11A). Baseline transcriptomic profiles were established from uninjured hearts to compare the cardiac transcriptomics of the wild-type control and *nr4a1* mutant. Given the crucial involvement of inflammation in heart regeneration, samples were also collected at 2 and 7 dpci, corresponding to stages marked by an active inflammatory response. Moreover, to elucidate the prolonged fibrosis phenotype in the late injury stage, we performed transcriptomic analysis on hearts collected at 21 dpci. Principal component analysis (PCA) demonstrated that biological replicates grouped closely together, indicating strong reproducibility within each experimental condition (Fig. 6A). Moreover, samples from different time points were clearly separated on the PCA plot, reflecting the distinct gene expression profiles associated with the

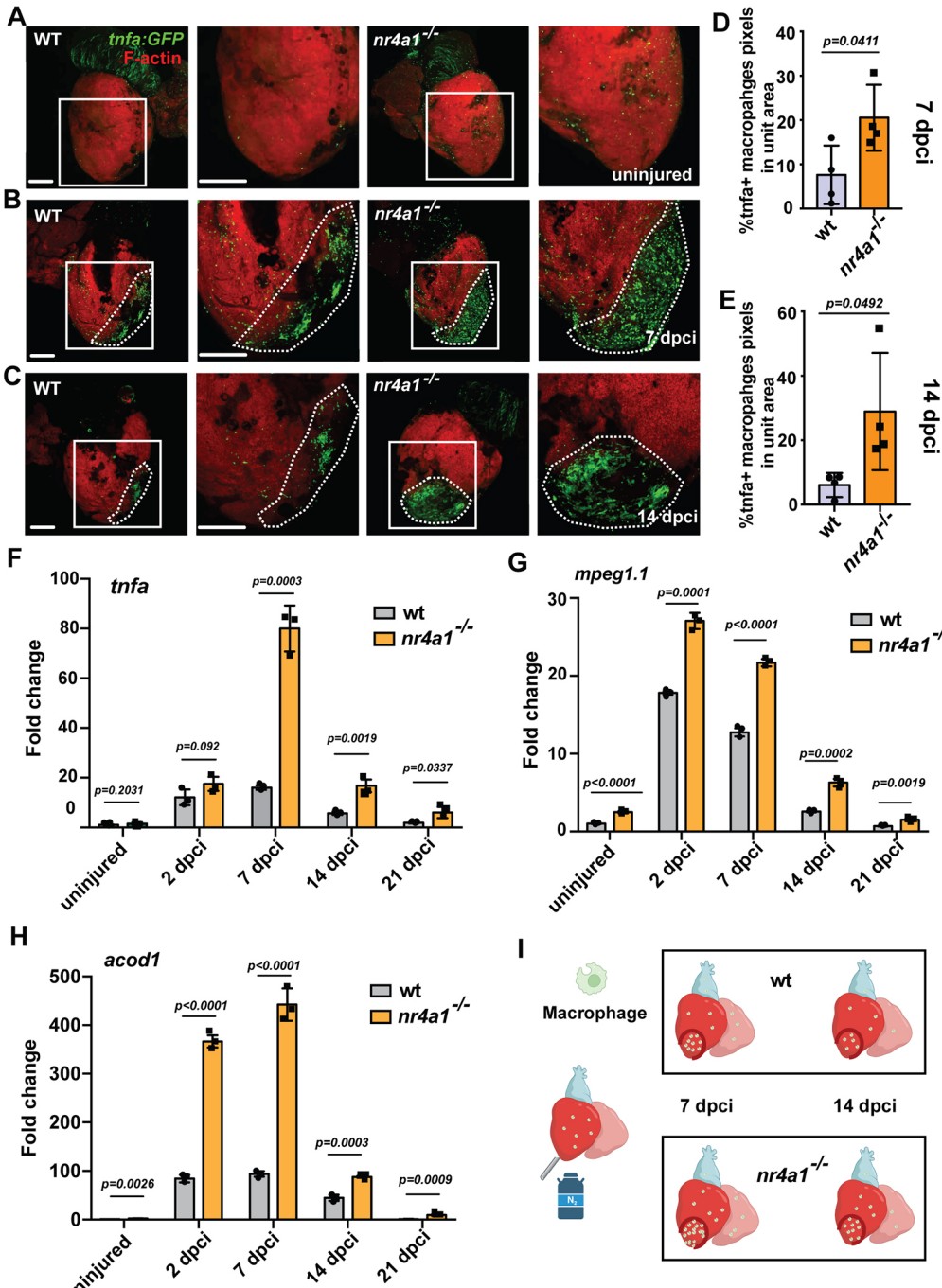

**Fig. 4. Abnormal accumulation of inflammatory macrophages in *nr4a1* mutant.** (A-C) Left: distribution of inflammatory macrophages in uninjured wild-type (WT) hearts and WT hearts at 7 and 14 dpci. Right: distribution of inflammatory macrophages in uninjured *nr4a1* mutant hearts and *nr4a1* mutant hearts at 7 and 14 dpci. Boxed regions in A-C show the approximate positions for quantification, and position of magnified images. Dashed lines show the wounded area. (D,E) Quantification of inflammatory macrophages in the unit area at 7 and 14 dpci. (F-H) qPCR results of inflammation-related genes *tnfa*, *mpeg1.1* and *acod1*. All fold changes are calculated relative to the uninjured WT group in each panel. (I) Schematic showing inflammatory macrophage distribution in WT and *nr4a1* mutant. Created in BioRender by Feng, D., 2025. https://BioRender.com/ynn17bt. This figure was sublicensed under CC-BY 4.0 terms. *P*-values<0.05 were considered statistically significant (two-tailed unpaired *t*-test, D,E; two-way ANOVA with Sidak test for multiple comparison correction and two-tailed unpaired *t*-test, F-H). Points in D and E show the individual samples. Data are mean±s.e.m. Scale bars: 275 μm.

various stages of cardiac injury (Fig. 6A). The first two principal components captured the primary variance related to temporal changes in gene expression following injury and between wild-type and mutant samples (Fig. 6A).

To identify significant changes in gene expression resulting from the loss of Nr4a1 function, we performed differential expression analysis between control and mutant samples at the same post-cardiac-injury stage. A log2 fold change greater than 0.5 and adjusted *P*-value less than 0.05 were used to define differentially expressed genes (DEGs). When comparing *nr4a1* mutant samples to controls, we identified 1079 upregulated and 1279 downregulated DEGs in the uninjured mutant ventricles (Fig. 6B-D), respectively. At 2 dpci, there were 780 upregulated and 782 downregulated DEGs

in the mutant group (Fig. 6B-D). At 7 dpci, 501 upregulated and 445 downregulated DEGs were observed in the mutants (Fig. 6B-D). At 21 dpci, we identified 686 upregulated and 754 downregulated DEGs in the mutants (Fig. 6B-D). Surprisingly, we observed more injury response-specific DEGs at 2 dpci and 21 dpci than at 7 dpci (Fig. 6B,C), suggesting dramatic alterations in the mutant hearts occurring at acute and late stages. Of note, we found that inflammation-related *acod1* remains increased after injury, and expression of matricellular gene *postnb* persists at 21 dpci in the non-regenerative mutant hearts (Fig. 6D). Subsequently, we performed hierarchical clustering of all DEGs based on their expression dynamics corresponding to time points and samples (Fig. 6E; Fig. S11B). Before injury, the Gene Ontology (GO) terms

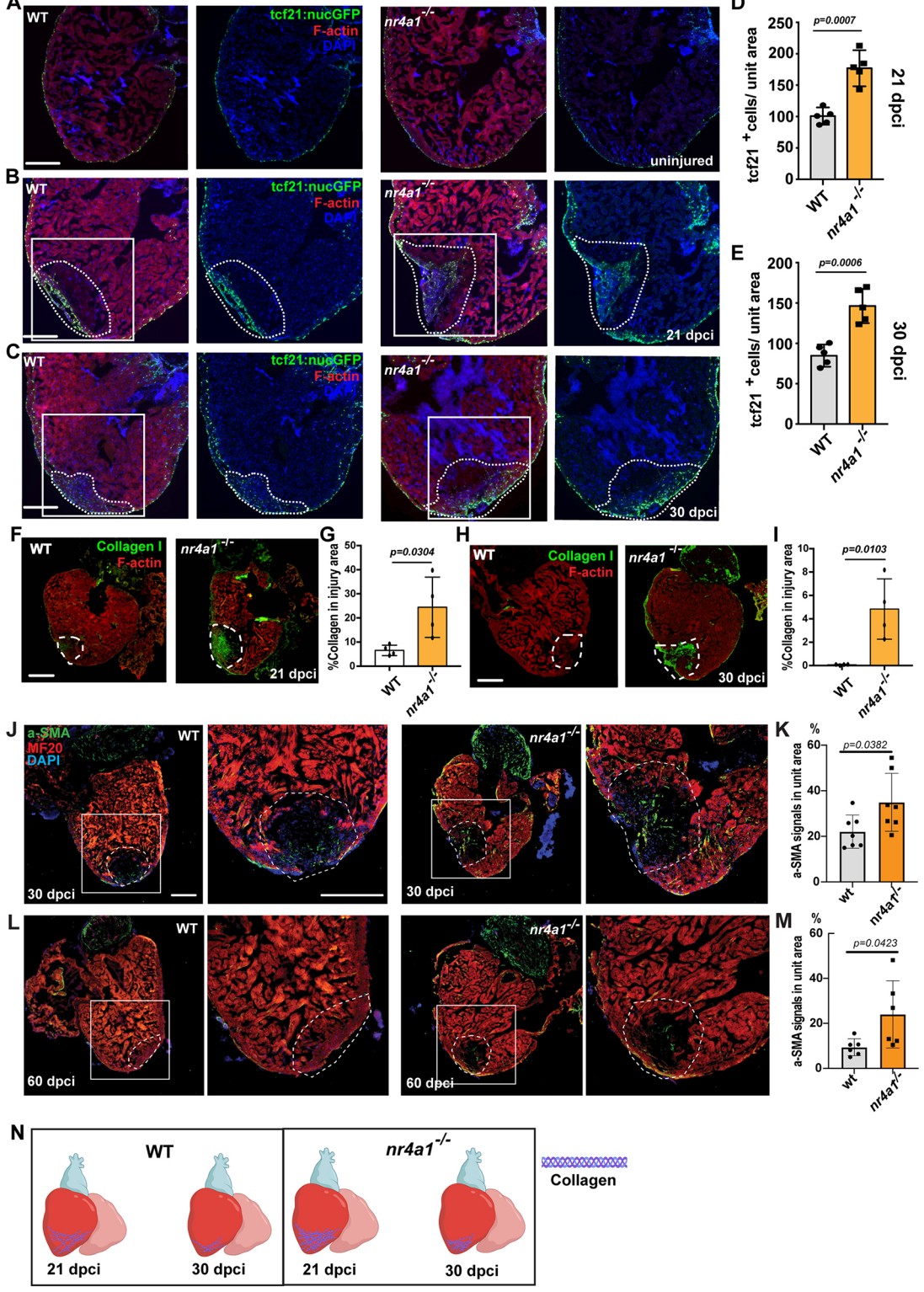

**Fig. 5. Persistent fibrosis in *nr4a1* mutant after cardiac injury.** (A-C) Left: distribution of fibroblasts on cardiac sections of uninjured wild-type (WT) hearts or WT hearts at 21 and 30 dpci. Right: distribution of fibroblasts on cardiac sections of uninjured *nr4a1* mutant hearts or *nr4a1* mutant hearts at 21 and 30 dpci. (D,E) Quantification of fibroblasts per unit area for WT and *nr4a1* mutant hearts at 21 and 30 dpci. (F,H) Collagen deposition, as revealed by collagen I antibody staining, in WT and *nr4a1* mutant at 21 dpci and 30 dpci, respectively. (G,I) Quantification of collagen deposition in the wounded area of WT and *nr4a1* mutant hearts at 21 dpci and 30 dpci, respectively. (J,L) α-SMA immunostaining at 30 dpci and 60 dpci injured hearts, respectively. Left: WT group; Right: *nr4a1* mutant group. (K,M) Quantification of α-SMA signal in the wounded area at 30 dpci and 60 dpci, respectively. (N) Schematic showing persistent fibrosis in *nr4a1* mutant. Dashed lines show the wounded area. Boxed regions show the approximate positions for quantification, and position of magnified images. Created in BioRender by Feng, D., 2025. https://BioRender.com/ok17j9e. This figure was sublicensed under CC-BY 4.0 terms. *P*-values<0.05 were considered statistically significant (two-way ANOVA with Sidak test for multiple comparison correction and two-tailed unpaired *t*-test). Points in D, E, G, I, K and M show the individual samples. Data are mean±s.e.m. Scale bars: 275 µm.

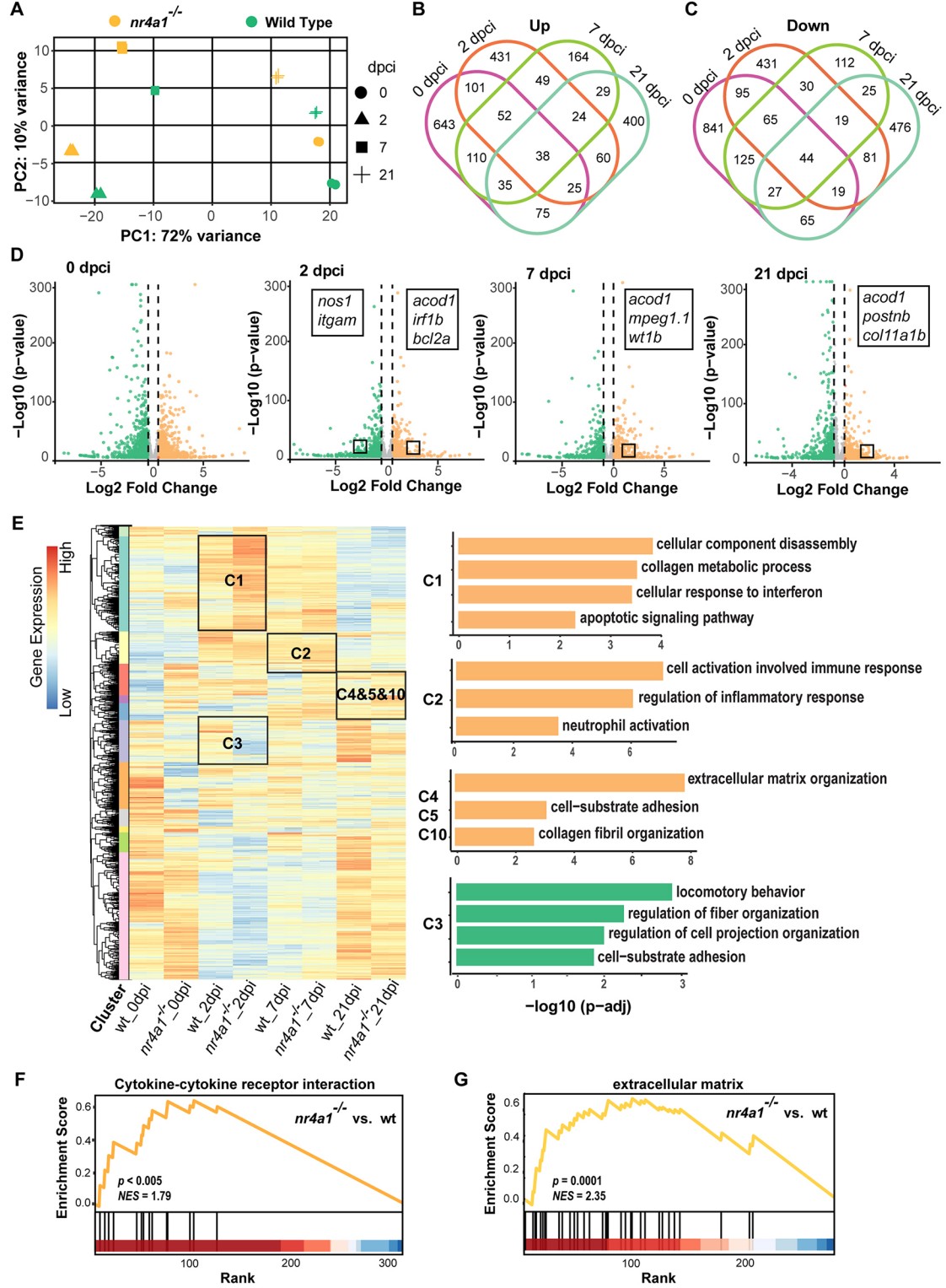

**Fig. 6. Upregulation of inflammation-, fibrosis- and apoptosis-related genes in *nr4a1* mutant revealed by transcriptomic analysis.** (A) Principal component analysis (PCA) plots represent the gene expression in hearts at different time points. (B,C) Venn diagrams of the upregulated (B) or downregulated (C) differentially expressed genes (DEGs) between control and mutant hearts from different stages after cryoinjury. (D) Volcano plots showing the DEGs at different stages after injury. Some representative genes are marked. Orange dots: upregulated DEGs in *nr4a1* mutant; green dots: downregulated DEGs in *nr4a1* mutant; log2 fold change ≥0.5; adjusted *P*-value<0.05. Color intensity represents the density of points in the volcano plot. (E) Hierarchical clustering heatmap of the comparatively DEGs in control and *nr4a1* mutant hearts. A total of 12 clusters were generated, and cluster-enriched genes involved in their predicted GO biological processes were listed in the right panels. (F) Gene set enrichment analysis (GSEA) plots of DEGs encoding cytokine-cytokine receptor interaction linked with ablation of *nr4a1* at 21 dpci. (G) GSEA plots of DEGs encoding ECM linked with deletion of *nr4a1*. The barcode plot indicates the positions of genes ranked by relevance, with red and blue colors denoting upregulation or downregulation, respectively, in the mutant fish hearts.

associated with the upregulated genes in the mutants are related to the regulation of translation and cellular respiration. In contrast, those associated with downregulated genes are mostly involved in multiple metabolic processes. In response to cardiac injury, a significant number of genes associated with lysosome organization, cell cycle activity and immune response were upregulated in 2 dpci control hearts (Fig. S11B, cluster 3 and cluster 2). In mutant hearts at 2 dpci, these upregulated genes were further accompanied by a pronounced enrichment of genes associated with cellular component disassembly, apoptotic signaling pathways and response to interferon (Fig. 6E, cluster 1). The enrichment of apoptotic processes in the mutants is consistent with the previously documented phagocytosis defects in macrophages and neutrophils observed in *nr4a1* mutants (Elliott and Ravichandran, 2010; Ankawa et al., 2021; Riwaldt et al., 2021). To experimentally validate this enrichment of apoptosis, we performed TUNEL staining, which revealed a significant accumulation of apoptotic cells in the injured areas of *nr4a1* mutants compared to controls (Fig. S12A,B). Notably, treatment with the immunomodulatory drug pomalidomide (Pom) reduced the number of apoptotic cells at the injury site in *nr4a1* mutants to levels comparable to those observed in wild-type controls at 7 dpci (Fig. S12A,B). To further dissect the relationship between inflammation and apoptosis in the *nr4a1* mutants, we inhibited apoptosis directly. We found that this had no significant effect on the inflammatory response in *nr4a1* mutant hearts (Fig. S12C,D). This finding suggests that excessive inflammation is responsible for the increased apoptosis observed in *nr4a1* mutants. In addition, DEGs associated with locomotory behavior, cell projection, cytoskeleton organization and cell-substrate adhesion were downregulated in the *nr4a1* mutant (Fig. 6E, cluster 3). For example, among DEGs of the 'locomotory behavior' ephrin type A receptor 4 (*epha4a*) and *cxcl12a* play essential roles in neutrophil infiltration and inflammatory response (Woodruff et al., 2016; Isles et al., 2019; Ryan et al., 2020), supporting our observation of disrupted neutrophil recruitment at the wound area in *nr4a1* mutants at the acute phase. Following 2 dpci, genes involved in inflammatory response gradually decreased in control wild-type hearts (Fig. S11B, cluster 2), whereas DEGs involved in ECM organization showed increased expression at 7 dpci and subsequently downregulated at 21 dpci (Fig. S11B, clusters 1 and 4). In contrast, the expression of the genes associated with GO terms related to immune response remained elevated in mutant hearts throughout the stages (Fig. 6E, cluster 2). Correspondingly, inflammation-related genes, including *il1b*, *tnfrsf1b*, *itgb2* and *irf8* (Herter and Zarbock, 2013; Mitroulis et al., 2015; Wajant and Siegmund, 2019; Denans et al., 2022), were persistently elevated in mutants compared with controls. Consistent with the observation of unresolved fibrosis at 21 dpci in the mutants (Fig. 6E, clusters 4, 5 and 10), genes associated with the GO term 'collagen fibril organization', including *col1a1a*, *col1a2*, *col2a1* and *mmp11a* were found to be upregulated in the mutant hearts at 21 dpci compared to the control hearts (Caldeira et al., 2018). The upregulation of collagen or inflammation-related genes was further validated by qRT-PCR (Fig. S13). Moreover, gene set enrichment analysis (GSEA) of the differentially expressed genes revealed significant enrichment for cytokine-cytokine receptor interaction and ECM-related process in the *nr4a1* mutant at 21 dpci (Fig. 6F,G). These findings are consistent with our previous observation of prolonged inflammation and persistent fibrotic scarring in the *nr4a1* mutant. Collectively, these findings suggest that *nr4a1* plays a crucial role in cardiac regeneration, primarily through the modulation of immune cell dynamics and the resolution of fibrosis.

## Inhibition of TNF signaling partially rescued heart regeneration in *nr4a1* mutants

Previous studies have demonstrated that *tnfa*, a crucial regulator of the inflammatory response following tissue damage, is indispensable for effective tissue regeneration in zebrafish (Dong et al., 2024). However, dysregulation of Tnfα signaling can lead to chronic inflammation and impair tissue regeneration (Bradley, 2008). Our finding that the loss of function of *nr4a1* significantly increased the expression of *tnfa* (Fig. 4F) indicates heightened inflammation (Liu et al., 2023). We, therefore, hypothesized that reducing Tnfα signaling to decrease the inflammatory level may enhance heart regeneration in *nr4a1* mutant hearts. To test this possibility, we generated the *nr4a1$^{-/-}$;tnfa$^{+/-}$* compound mutant and performed cryoinjury on both the compound mutants and the control *nr4a1* mutant hearts. AFOG staining showed that the scar area in the compound mutant was significantly reduced compared to the control *nr4a1* mutants at 60 dpci (Fig. 7A,B), indicating that reducing Tnfα signaling could improve cardiac regeneration in the *nr4a1* mutant. Consistently, compared to the *nr4a1* mutant, the hearts of the compound mutants had significantly reduced collagen deposition (Fig. 7C,D), indicating that reducing Tnfα signaling could reduce cardiac fibrosis in the *nr4a1* mutant. To further examine heart regeneration in the compound mutants, we performed the CM proliferation assay by PCNA/Nkx2.5 double antibody staining. We observed a significantly higher number of proliferating CMs in double mutants than in *nr4a1* mutants, with no significant difference with wild-type control (Fig. 7E,F). Meanwhile, immunostaining against Mpx and IB4 showed that hearts of the *nr4a1$^{-/-}$tnfa$^{+/-}$* compound mutants had significantly reduced number of neutrophils and macrophages compared to the *nr4a1* mutant at 7 dpci and 14 dpci, which is consistent with the decreased inflammatory level in the compound mutants (Fig. S14). Additionally, we assessed the expression level of inflammation-related and fibrosis-related genes between the compound and control mutant hearts. We found a significantly reduced level of *tnfa*, *il1b* and *acod1* in *nr4a1$^{-/-}$;tnfa$^{+/-}$* double mutants (Fig. 7G). Similarly, the expression of *postnb*, *fn1a* and *col12a1a* levels also exhibited a downward trend in the compound mutant hearts compared to the mutant hearts (Fig. 7H). In summary, all these results indicate that reducing Tnfα signaling can significantly lower the inflammation level, thereby improving the heart regeneration in *nr4a1* mutants that exhibit chronic inflammation.

## DISCUSSION

In this study, we found that loss of Nr4a1 function results in heightened chronic inflammation after cardiac injury and compromises cardiac regeneration. Specifically, we observed an increased recruitment of inflammatory macrophages to the wounded area, with a substantial presence of these macrophages persisting into the late stages of injury. Furthermore, transcriptome-wide bulk RNA-seq analysis indicates that *nr4a1* mutants demonstrate enhanced regulation of inflammatory response and cellular response to interferon. Consistent with these findings, we observed upregulation of inflammatory-related genes, including *tnfa*, in *nr4a1* mutant hearts compared to controls. Tnfα is the widely acknowledged pro-inflammatory cytokine that is integral to the regulation and propagation of the inflammatory response (Bradley, 2008). Our study aligns with previous studies in mice, which show that the loss of Nr4a1 function significantly elevates *Tnfa* expression in a mouse model of neuronal injury (Liu et al., 2023). Although recent studies suggest that inflammation is necessary for tissue repair, with Tnfα signaling crucial for heart regeneration in zebrafish (Dong et al., 2024), we hypothesize that a chronically heightened inflammatory

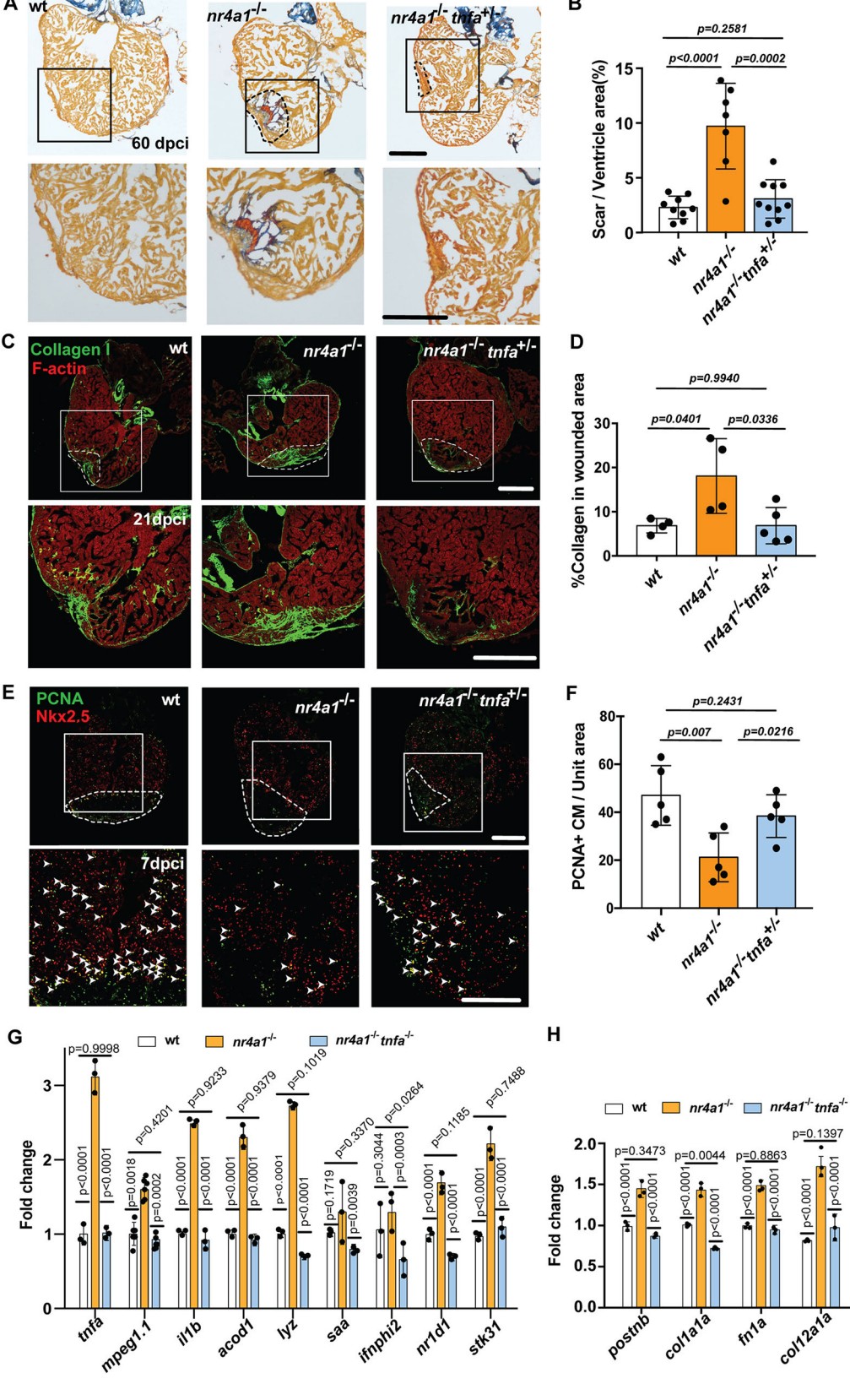

**Fig. 7. Partial restoration of cardiac regeneration in *nr4a1* mutant through reduction of inflammatory response.** (A) AFOG staining performed on cardiac sections of the indicated genotypes at 60 dpci. (B) Quantification of scar size in AFOG-stained cardiac sections. (C) Collagen deposition in the hearts of indicated genotypes at 21 dpci, as detected by staining with collagen I antibody. (D) Quantification of collagen deposition in the wounded area at 21 dpci. (E) PCNA/Nkx2.5 double immunostaining to detect proliferating cardiomyocytes in 7 dpci hearts of indicated genotypes. Arrowheads indicate the proliferating cardiomyocytes. (F) Quantification of colocalization of PCNA and Nkx2.5 per unit area in different groups. (G) qPCR of inflammation-related genes in the hearts of indicated genotypes at 7 dpci. (H) qPCR of fibrosis-related genes in the hearts of indicated genotypes at 7 dpci. Dashed lines show the wounded area. Boxed regions in A and E show the position of magnified images at the bottom. Boxed regions in C show the approximate positions for quantification. The fold changes in G and H are calculated relative to the expression in uninjured WT hearts. *P*-values<0.05 were considered to be statistically significant (two-tailed unpaired *t*-test in B, D and F; two-way ANOVA with Sidak test for multiple comparison correction and two-tailed unpaired *t*-test in G and H). Data are mean±s.e.m. Scale bars: 275 µm (A, E); 250 µm (C).

response could impair tissue repair and regeneration. Indeed, reducing Tnfα signaling in *nr4a1* mutants decreased cardiac fibrosis and improved cardiac regeneration. These findings highlight the delicate balance required for optimal inflammatory responses in tissue repair and suggest that modulating Tnfα signaling may be a therapeutic strategy for enhancing cardiac regeneration in chronic inflammation.

Neutrophils are essential first responders to injury, rapidly accumulating at the site of damage and initiating the inflammatory

response (Kolaczkowska and Kubes, 2013). Our results reveal an aberrant distribution of neutrophils during heart regeneration in *nr4a1* mutants, characterized by delayed accumulation at the site of injury in the early stages, followed by a persistent and substantial presence during the later stages of injury. These observations suggest a dysregulation of neutrophil function in *nr4a1* mutants, potentially contributing to impaired heart regeneration in mutant hearts. It is well established that the efficient clearance of cellular debris by neutrophils is essential for resolving inflammation and promoting tissue repair (Herrero-Cervera et al., 2022). Failure to effectively clear dead cell debris and pathogens can exacerbate inflammation, thereby impeding the healing process (Maderna and Godson, 2003; Westman et al., 2020). Our observation of the enrichment of GO terms related to apoptotic pathways suggests that the dysfunctional neutrophils in *nr4a1* mutants may fail to effectively clear apoptotic cell debris, further exacerbating the inflammatory response. While a recent study has reported that NR4A2 and NR4A3, other members of the NR4A family, are essential for neutrophil production, more evidence is needed to establish the relationship between Nr4a1 and neutrophil function (Prince et al., 2017).

Our study also revealed a dysregulated fibrotic response in *nr4a1* mutants, particularly at the late injury stages. Transient fibrosis is a crucial component of zebrafish heart regeneration (Sánchez-Iranzo et al., 2018). Most recently, the importance of precisely balancing pro-regenerative and pro-fibrotic ECM gene expression has been highlighted in zebrafish heart regeneration (Marro et al., 2016; Allanki et al., 2021; Hu et al., 2022; Izu and Birk, 2023). While some key pro-regenerative ECM genes are upregulated in *nr4a1* mutants, their prolonged expression, combined with elevated pro-fibrotic ECM gene expression, likely impair fibrosis resolution and promote fibrotic scarring in the mutants. Previous studies have demonstrated the role of inflammatory macrophages in the activation of fibroblasts, with the inhibition of pro-inflammatory macrophage function significantly reducing fibroblast accumulation at the injury site (Ma et al., 2021). Therefore, the sustained activation of inflammatory macrophages in *nr4a1* mutants may contribute to the persistence of activated fibroblasts in the wounded area of the heart, potentially exacerbating fibrosis and impairing tissue repair.

At the transcriptome level, we identified many GO terms related to inflammation and fibrosis. Following injury, genes associated with inflammatory response and leukocyte migration are significantly upregulated in *nr4a1* mutants. These GO terms highlight altered macrophage and neutrophil functions in *nr4a1* mutants. We identified GO terms associated with the apoptotic process, a finding further corroborated through Tunnel assays, which revealed an increased presence of apoptotic cells in *nr4a1* mutants. qRT-PCR analysis of apoptotic-related genes demonstrated a consistent upregulation of these genes, reinforcing these observations. Macrophages and neutrophils play a crucial role in phagocytosis of cell debris during the resolution of inflammation (Nauseef and Borregaard, 2014; Westman et al., 2020). One plausible explanation for the impaired resolution of inflammation in *nr4a1* mutants is the dysfunction of macrophages and neutrophils in their ability to effectively engulf apoptotic cells and debris, thereby disrupting the normal inflammatory resolution process.

Taken together, our study demonstrates that loss of Nr4a1 function leads to significant disruptions in the inflammatory response and tissue regeneration processes. We observed aberrant neutrophil and macrophage activity, characterized by delayed and prolonged inflammatory responses and impaired clearance of apoptotic cells. These dysregulated immune responses contribute to chronic inflammation and persistent fibrosis, ultimately impeding effective heart regeneration. Our findings underscore the essential role of Nr4a1 in regulating immune cell function during tissue repair and highlight its potential as a therapeutic target for mitigating chronic inflammation and enhancing regenerative outcomes.

## MATERIALS AND METHODS

### Zebrafish lines and husbandry

Zebrafish husbandry and experiments were conducted following the University of North Carolina at Chapel Hill Institutional Animal Care and Use Committee (IACUC). Zebrafish were maintained under standard conditions at 28.5°C on a 14 h light/10 h dark cycle. Embryos and larvae were raised in 0.5× E2 solution (7.5 mM NaCl, 0.25 mM KCl, 0.5 mM MgSO$_4$, 75 µM KH$_2$PO$_4$, 25 µM Na$_2$HPO$_4$, 0.5 mM CaCl$_2$, 0.35 mM NaHCO$_3$, 0.5 mg/l Methylene Blue). Adult zebrafish used in the study were between 0.5 to 1 year old.

The *tnfa$^{-/-}$ (sa43296)* mutant line was obtained from the Zebrafish International Resource Center (ZIRC). Previously generated transgenic lines used in this study are *TgBAC(tnfa:eGFP)* (Marjoram et al., 2015), *Tg(mpeg1: eGFP)* (Ellett et al., 2011), *Tg(lyz:eGFP)* (Hall et al., 2007), *TgBAC(tcf21: nuc-eGFP)* (Wang et al., 2011). The *nr4a1$^{-/-}$ (nc40)* mutant was generated using CRISPR/Cas9. sgRNAs were designed using the online CRISPR-Cas9 guide RNA design tool developed by Integrated DNA Technologies (IDT). sgRNA (5-GGTAGCAGCCATACACCTGG-3) were synthesized by using Invitrogen™ MEGAscript™ T7 Transcription Kit. We injected 2 nl of a RNP mix of Cas9 protein (IDT, 150 ng/µl) with the sgRNA (70 ng/µl) into one-cell-stage embryos. We identified the founder fish with 1 bp insertion. Genotyping was performed (using primers: Fwd, 5-CACCTACTCATGC-CAGATCG-3; Rv, 5-CTGATCTGGGATCTGTGACTGC-3) followed by enzyme digestion BstXI (New England Biolabs).

### KI transgenic line generation

The KI transgenic line was generated using the recently published nonhomologous end joining (NHEJ) repair method (Levic et al., 2021). We used CRISPR-Cas9 to induce the double-strand DNA (dsDNA) break in the last intron of the *nr4a1* genome with the sgRNA 5-TGCGTTGGATTCAACAATTG-3. The plasmid for DNA donor construction was purchased from Addgene (plasmid #174024). For 5′ integration detection, genotyping primers were: Fwd, 5-GGAAGCCTACTATGT-TAGGTG-3; Rv (*eGFP*), 5-GTAGGTCAGGGTGGTCACG-3. For 3′ integration detection, genotyping primers were: Fwd (*eGFP*), 5-GATCCGC-CACAACATCGAG-3; Rv, 5-TGTCGATCTGTTGGTCACAG-3.

### Cryoinjury

Adult zebrafish were anesthetized in a 0.168 mg/ml Tricaine solution (Sigma-Aldrich, MS222). The fish were then positioned on a water-soaked sponge containing a small groove to ensure stability. A small incision was made in the body wall and the pericardium to expose the heart. The ventricular apex was then touched with the tip of a steel probe, pre-cooled by immersion in liquid nitrogen, and held in contact until thawing occurred (~7-10 s). Following this, the fish were transferred into the system water and awakened by continuous pipetting (González-Rosa and Mercader, 2012).

### Heart dissection and cryosection

Adult zebrafish were euthanized by immersion in ice water for at least 15 min to induce hypothermia. The hearts were subsequently harvested, following decapitation, from the base of the pectoral fin using a razor blade and then fixed in 4% paraformaldehyde (PFA) at 4°C overnight. After washing with PBS containing 0.1% (v/v) Tween-20 (0.1% PBST), hearts were either stored in methanol at −20°C for long-term storage or maintained in 0.1% PBST overnight for cryosection and staining. For cryosectioning, hearts were dehydrated in a 30% sucrose solution (W/V in PBS) overnight at 4°C to ensure cryopreservation. The hearts were then embedded in molds filled with OCT (Tissue-Tek) and stored at −80°C. Cryosections of 10 µm thickness were collected from the frozen hearts using a Leica CM 1950 for subsequent histological analysis and immunofluorescent staining.

## AFOG staining

Cryosections were initially fixed in Bouin's solution for 2 h at 60°C, then allowed to cool down to room temperature (RT). The slides were rinsed with running tap water for 30 min. The tissues were treated with 1% phosphomolybdic acid and 0.25% phosphotungstic acid solution for 5 min, followed by a 5-min wash with distilled water. Subsequently, the sections were incubated for 5 min with AFOG solution, which contains 1 g Aniline Blue (Santa Cruz Biotechnology), 3 g acid Fuchsin (Roth), 2 g Orange G (Santa Cruz Biotechnology) in distilled water, with pH adjusted to 1.1. After another 5-min wash with distilled water, the sections were dehydrated through two changes of absolute ethanol and two changes of xylene, then mounted with Permount mounting medium (Fisher Chemical).

## RNA extraction and real-time PCR

Total RNA was isolated from zebrafish ventricles using Trizol (Invitrogen). Reverse transcription was performed using SuperScript IV VILO Master Mix (Invitrogen). Real-time PCR was performed with SYBR Green (Applied Biosystems) on QuantStudio Flex 6 (Applied Biosystems). *elfa* was used as the reference gene expression using the ΔΔCt method. The primers used are listed in Table S1. All data were processed with three biological replicates.

## Immunofluorescence and histological staining

For immunostaining of cardiac cryosections, the sections were boiled in citrate buffer (10 mM, pH 6.0) for 30 min at 96°C, then allowed to cool down to RT. The sections were washed three times with 0.1% PBST for 5 min each. Tissues were then blocked for 1 h with a blocking buffer containing 10% (v/v) sheep serum in PBS. Primary antibodies, diluted in the blocking solution, were applied to the sections and incubated overnight at 4°C. The following day, sections were washed four times with 0.1% PBST. Secondary antibodies and Phalloidin were added to the sections and incubated at RT for 4 h, followed by four 0.1% PBST washes. If needed, DAPI (Sigma-Aldrich) was added for a 2-min incubation after the final wash. Finally, sections were mounted with ProLong™ Diamond Antifade Mountant (Invitrogen). The following primary antibodies were used: rabbit anti-GFP (1:500, Invitrogen, A-11122), rabbit anti-Mpx (1:200, GeneTex, GTX128379) and rabbit anti-Collagen I (1:200, Abcam, ab138492). The following secondary antibodies were used: Alexa Fluor 647 anti-rabbit (1:500, Thermo Fisher Scientific, A-21245), Alexa Fluor 568 Phalloidin (1:500, Invitrogen, A12380). For IB4 staining to mark the macrophage (1:100, Griffonia Simplicifolia Lectin I Isolectin B4, Vector Laboratories, DL-1207-.5), the IB4 dye was added in the blocking buffer along with secondary antibody during immunostaining of the sections.

For immunostaining of whole-mount heart samples, the samples were first fixed in 4% PFA overnight and then boiled in citrate buffer (10 mM, pH 6.0) for 30 min at 65°C. The hearts were washed four times with 0.5% PBST for 5 min each. The samples were then transferred into blocking buffer (5% sheep serum, 1% DMSO in 0.5% PBST, v/v) for 2 h at RT. After primary antibody incubation, the samples were washed four times with 0.5% PBST for 5 min each and then incubated with the secondary antibody at 4°C overnight. The hearts were washed again four times with 0.5% PBST for 5 min each and then incubated in ScaleA2 solution at 4°C for 2 days to perform tissue clearing. Finally, the heart samples were mounted in low melting point agarose (Sigma-Aldrich) for imaging.

For cell proliferation staining using PCNA, tissue sections were initially subjected to antigen retrieval by boiling in citrate buffer (10 mM, pH 6.0) for 30 min at 96°C. After cooling down to RT, the sections were washed four times with 0.1% PBST for 5 min each. The sections were then blocked with the blocking buffer containing 10% sheep serum (v/v) in 0.1% PBST at 37°C for 30 min. Primary antibody incubation was performed using a mouse anti-PCNA antibody (1:200, Invitrogen, 13-3900) diluted in the blocking buffer with 10% sheep serum in 0.1% PBST at 37°C for 3 h. After incubation, the slides were washed three times with 0.1% PBST, each for 5 min. Subsequently, the sections were incubated overnight at RT with rabbit anti-Nkx2.5 antibody (1:200, GeneTex, GTX128357) diluted in the blocking buffer. The following day, the slides were washed four times with 0.1% PBST, each for 5 min. Secondary antibody incubation was carried out with Alexa Fluor 647 anti-rabbit (1:500, Thermo Fisher Scientific, A-21245) and Alexa Fluor Cy3 anti-mouse (1:500, Thermo Fisher

Scientific, A10520) antibodies for 4 h at RT. Finally, the sections were washed four times with 0.1% PBST, and the slides were mounted using ProLong™ Diamond Antifade Mountant (Invitrogen, P36935) for imaging. The TUNEL assay was conducted to detect apoptotic cells using the *In Situ* Cell Death Detection Kit, Fluorescein (Roche, 11684795910), following the manufacturer's protocol.

## Imaging

Imaging was performed using the Zeiss LSM 800 confocal microscope and an Invitrogen EVOS M7000. A Zeiss AXIO Zoom V16 microscope was used for AFOG staining imaging.

## *In situ* hybridization

The *in situ* probe for *postnb* was synthesized by first amplifying the DNA fragment using the following primers: Fwd, 5-GCTCCTACCGATGA-AGCTTTCG-3; Rv, 5-CTAGGTTCTCCTCCGGACG-3. The probe synthesis was conducted using the Invitrogen™ MEGAscript™ T7 Transcription Kit with the DIG RNA labeling mix (Roche). We followed the protocol described on the Abcam website (https://www.abcam.com/protocols/ish-in-situ-hybridization-protocol). Gene expressions were detected with the anti-DIG AP, Fab fragment (Roche) and NBT-BCIP substrate system (Sigma-Aldrich).

## RNAscope fluorescence *in situ* hybridization

After fixation, the cardiac tissues were embedded in OCT compound, and 10-μm sections were prepared using a Leica CM1950 cryostat. RNAscope *in situ* hybridization was performed according to the manufacturer's protocol using the Multiplex Fluorescent V2 Assay (Advanced Cell Diagnostics, 323110). For detection, the TSA Plus Cyanine 5 System (Perkin Elmer, NEL745001KT) was used. Slides were covered using Fluor-Gel II with 4′,6-diamidino-2-phenylindole (Electron Microscopy Sciences, 17985-50).

## Pomalidomide and apoptosis inhibitor treatment

Pharmacological inhibitor of Tnfα release, pomalidomide (5 μl at a concentration of 10 mg/ml; Cayman Chemicals, 19877), apoptosis inhibitor (2 μl at a concentration of 10 mg/ml, Sigma-Aldrich, 178488) or PBS were injected intrathoracically into individual anaesthetized zebrafish 1 day before cryoinjury using a Picospritzer microinjector (Parker). The injection was repeated one more time at 2 dpci.

## Western blot

Western blotting was performed according to standard protocol with rabbit anti-GFP (1:500, Invitrogen, A-11122) as the primary antibody.

## Bulk RNA-seq and analysis

Total RNA was extracted from zebrafish according to the manufacturer's protocol. For RNA-seq, library preparation and Illumina RNA-seq were performed by Novogene. RNA-seq raw data quality was checked by FastQC (https://www.bioinformatics.babraham.ac.uk/projects/fastqc/). RNA-seq were aligned to the zebrafish reference genome using STAR (Dobin et al., 2013). Gene expression was quantified as gene hit counts (reads per gene) using FeatureCounts (Liao et al., 2014). To visualize the global structure of the data and assess the impact of batch and condition effects, a PCA plot was generated based on the log-transformed count data to explore the variance attributed to different conditions and time points using ggplot2 (https://ggplot2.tidyverse.org/). DEGs between time points (0, 2, 7 and 21 dpci) and condition (wild type or mutant) were identified using DESeq2 ($P<0.05$) (Love et al., 2014), while accounting for batch effects. Unsupervised hierarchical clustering was performed to analyze gene expression patterns. Gene clusters that were differentially expressed were visualized in heatmaps using the pheatmap package. GO enrichment and GSEA were conducted for each cluster using clusterProfiler (Wu et al., 2021). GO terms with an adjusted *P*-value of <0.05 were considered significant.

## Statistical analysis

All data are presented as mean±s.e.m. and were statistically analyzed using Prism software (GraphPad Prism 10). When comparing data from more than

two groups, statistical significance was calculated using two-way ANOVA with the Sidak test for multiple comparison correction. Data from two groups were compared using a two-tailed unpaired *t*-test. *P*-values smaller than 0.05 were considered to be statistically significant. Plot or bar graphs were made in Prism software. More details about the quantification methods are listed in the Supplementary Materials and Methods.

## Acknowledgements
We thank the UNC Olympus Imaging Research Center for confocal use and the UNC Zebrafish Aquaculture Core Facility for fish care and microscope use.

## Competing interests
The authors declare no competing or financial interests.

## Author contributions
Conceptualization: J.L.; Formal analysis: D.F., Y.D., Y.S., L.K., N.Y.; Funding acquisition: J.L., L.Q.; Investigation: J.L., D.F., Y.D., T.L.; Methodology: J.L., D.F., Y.D.; Project administration: J.L., L.Q.; Resources: Y.X., B.S.; Supervision: J.L., L.Q.; Validation: Y.D.; Writing – original draft: J.L., D.F., Y.D., L.Q.; Writing – review & editing: J.L., Y.X., B.S., L.Q.

## Funding
This study was supported by the National Institutes of Health/National Heart, Lung, and Blood Institute (NIH/NHLBI) R01HL164933, R01HL168285 and R01HL174774 grants, the American Heart Association (AHA) grant 20EIA35320128 to J.L., and AHA 18TPA34180058, 20EIA35310348, and NIH R35HL15565656 grants to L.Q. Open Access funding provided by University of North Carolina, Chapel Hill. Deposited in PMC for immediate release.

## Data and resource availability
All sequencing data generated in this study have been deposited in GEO under accession GSE276850.

## Peer review history
The peer review history is available online at https://journals.biologists.com/dev/lookup/doi/10.1242/dev.204395.reviewer-comments.pdf

## Special Issue
This article is part of the Special Issue 'Lifelong Development: the Maintenance, Regeneration and Plasticity of Tissues', edited by Meritxell Huch and Mansi Srivastava. See related articles at https://journals.biologists.com/dev/issue/152/20.

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
