## [Peer Review File · Development (Cambridge, England)]

Nr4a1 modulates inflammation and heart regeneration in zebrafish

Dong Feng, Yanhan Dong, Yiran Song, Nicholas Yapundichl, Yifang Xie, Brian Spurlock, Tingting Lyu, Landry Kuehn, Li Qian and Jiandong Liu
DOI: 10.1242/dev.204395

Editor: Ken Poss

Review timeline

Original submission:	13 September 2024
Editorial decision:	26 October 2024
First revision received:	26 March 2025
Editorial decision:	20 April 2025
Second revision received:	1 June 2025
Accepted:	12 June 2025

Original submission

First decision letter

MS ID#: dev.204395

MS TITLE: Nr4a1 Modulates Inflammation and Heart Regeneration in Zebrafish

AUTHORS: Jiandong Liu; Dong Feng; Yanhan Dong; Yiran Song; Yifang Xie; Brian Spurlock; Landry Kuehn; Nicholas Yapundichl; Li Qian

Dear Jiandong,

I have now received all the referees' reports on the above manuscript, and have reached a decision. The referees' comments are appended below, or you can access them online: please go to:

As you will see, the referees express considerable interest in your work, but have some significant criticisms and recommend a substantial revision of your manuscript before we can consider publication. If you are able to revise the manuscript along the lines suggested, which may involve further experiments, I will be happy receive a revised version of the manuscript. Your revised paper will be re-reviewed by one or more of the original referees, and acceptance of your manuscript will depend on your addressing satisfactorily the reviewers' major concerns. Please also note that Development will normally permit only one round of major revision. If it would be helpful, you are welcome to contact us to discuss your revision in greater detail. Please send us a point-by-point response indicating your plans for addressing the referees' comments, and we will look over this and provide further guidance.

Please attend to all of the reviewers' comments and ensure that you clearly highlight all changes made in the revised manuscript. Please avoid using 'Tracked changes' in Word files as these are lost in PDF conversion. I should be grateful if you would also provide a point-by-point response detailing how you have dealt with the points raised by the reviewers in the 'Response to Reviewers' box. If you do not agree with any of their criticisms or suggestions please explain clearly why this is so.

Reviewer 1

Summary: Feng et al. explored the mechanism of adult heart regeneration, which they determined the role of Nr4a1 during the time course of inflammatory and regeneration to zebrafish cardiac cryoinjury. They first identified the dynamic expression of Nr4a1 in macrophages and neutrophils upon heart injury. Then they created zebrafish Nr4a1 mutant and these animals showed a prolonged and excessive inflammatory response, disrupted neutrophil migration, delayed fibrin clearance, and ultimately impaired heart regeneration. Further, they performed RNA-seq analysis at different stages and found the molecular disruptions associated with dysregulated inflammation and fibrosis in Nr4a1 mutants. Moreover, partial inhibition of Tnf $\hat{\pm}$ signaling rescued heart regeneration in the Nr4a1 mutants. The experiments are generally well performed and the manuscript is well written. The results are interesting, as understanding the mechanism of inflammation regulation during heart regeneration is critical for developing potential therapeutic strategies. There are several points that should be addressed in order to reinforce the conclusions proposed by the authors.

Concerns:

1. The authors showed the Nr4a1 expression in macrophages and neutrophils with knock-in reporter. It will be beneficial to examine/verify whether Nr4a1 is expressed in other cell types with published scRNA-seq datasets of zebrafish injury/regeneration heart.
2. They examined PCNA/CM at 7 dpi and scar formation in regenerated heart of Nr4a1 mutant. How about the myocardial restoration in mutant at 30-60 dpi?
3. The Nr4a1 is dynamically expressed during heart regeneration. Is this expression pattern correlated with inflammatory/regeneration events at different stages of heart recovery?
4. The authors stated that "significantly higher expression levels of the inflammatory-related genes tnfa...". Please clarify whether there are more tnfa⁺ cells or enhanced tnfa mRNA level in tnfa⁺ cells.
5. What's the effects of tnfa⁺/ - rescue on neutrophil and macrophage responses in Nr4a1^{-/-} regenerating hearts?

Reviewer 2

SUMMARY OF THE ADVANCE MADE IN THIS PAPER AND ITS POTENTIAL SIGNIFICANCE TO THE FIELD

Identified nuclear receptor Nr4a1 as a key regulator in zebrafish heart regeneration.

Nr4a1 deficiency led to:

- * Prolonged and excessive inflammation.
- * Disrupted neutrophil migration.
- * Delayed fibrin clearance.
- * Impaired heart regeneration.

RNA-seq analysis across injury stages revealed:

- * Molecular disruptions linked to unchecked inflammation and fibrosis in Nr4a1 mutants. Inhibiting pro-inflammatory cytokine Tnf- $\hat{\pm}$ partially restored heart regeneration in Nr4a1 mutants. These findings suggest Nr4a1 is crucial for managing the immune response during heart repair and may be a promising target for therapies aimed at enhancing cardiac regeneration after injury as such this study is novel and of significant interest to audience of Development.

Most of the central findings are supported by the data and the data is generally of sufficient quality for these conclusions to be made. I support publication of the manuscript with the following suggestions:

SUGGESTIONS TO AUTHORS

Major concerns

Figure 1: Uninjured control sections are required of the nr4a1KI line. Expression appears to rise after 7dpci, later RT-PCR timepoints are required to determine if/when this returns to uninjured levels. A similar time course of sections comparable to that of mpx and IB4 (Fig S1 and S2) would be helpful for understanding the dynamics of nr4a1 relative to these two cell types of interest.

Figure 1: Quantification would be helpful to better interpret later results: relative number of neutrophils and macrophage that are positive/negative for nr4a1+. Triple labeling should be used to determine if nr4a1 is only expressed by neutrophils or macrophage after injury.

Line 149/Figure 4: "A substantial accumulation of inflammatory macrophages was observed at the injury area of the control hearts at 7 dpci." This statement is not well supported without the use of definitive macrophage marker co-labeling with tnfa. Other cells, including activated neutrophils, have been reported to express tnfa.

Line 327: "Therefore, the sustained activation of inflammatory macrophages in nr4a1 mutants may contribute to the persistence of activated fibroblasts in the wounded area of the heart, potentially exacerbating fibrosis and impairing tissue repair." Again, there is a lack of data showing that activated/tnfa+ cells are macrophage in support of this statement. Further characterization of the tnfa+ cells is required.

Figure 7G: There is an inconsistency with the relative mpeg1.1 gene expression level in nr4a1 mutants between Figure 7G (not changed) and Figure 4G (significantly increased). This should be resolved.

Line 278/Figure 7I: col12a1a is included in the analysis but not mentioned in the text (also in Figure S7). This is not a recognized active fibroblast marker and may be pro-regenerative. Comments on the significance and interpretation of these result should be included.

Minor concerns

Line 63/64: Reference(s) required.

Line 186: "Collectively, our data indicated that nr4a1 mutant hearts exhibit impaired resolution of fibrosis, which is associated with compromised myocardial regeneration." It should be more clearly worded to better reflect that the associated data suggest that there is increased fibrotic accumulation rather than defective or slower breakdown of fibrosis.

Line 235: cxcl12a or b should be specified, only cxcl12a is known to signal to neutrophils in zebrafish.

Line 430-435 and 452-458 etc: There are multiple antibodies for these proteins/antigens available from these companies. Catalogue numbers or other unique identifiers are required to indicate which was used in this study.

Reviewer 3

Summary

In this study, Dong Feng et al. examine the nuclear receptor Nr4a1's involvement in heart regeneration in zebrafish. They find that nr4a1, a nuclear receptor encoding gene belonging to the Nr4a family, is upregulated after cardiac cryoinjury in zebrafish. Using transgenic approaches, the authors show a dynamic expression behavior of nr4a1 after injury, and show that nr4a1 is predominantly expressed in immune cells, specifically macrophages and neutrophils. To study Nr4a1 function, the authors establish new loss-of-function alleles for nr4a1 in zebrafish and find that these mutants show increased and extended signs of inflammation, along with a impaired heart regeneration. Additionally, transcriptome-wide RNA sequencing revealed that loss of Nr4a1 leads to broad immune response alterations, especially affecting neutrophils and macrophages. Neutrophil migration and clearance were disrupted in nr4a1 mutants, resulting in persistent inflammation. Excessive fibrosis and limited progression of regeneration were also observed. Interestingly, the authors found that partial inhibition of the cytokine Tnf- α could partially rescue the heart regeneration phenotype in nr4a1 mutants.

The work is of high quality and interesting to a broad range of scientists working in the field. Overall, it aligns well with previous findings on Nr4a1 in other model organisms and contributes to the identification of Nr4a1 function in regenerative biology. I did not identify major critical

problems with the presented results or the general interpretation. However, some aspects appear unclear and will need further experimental investigations, particularly those regarding fibrosis and apoptosis.

Comments:

1. The authors show that nr4a1 mutants present strongly elevated levels of apoptotic cells 7 days post-injury (Fig S9). Since NR4A1 has been directly linked to apoptosis in other systems, including mammalian models, it remains unclear whether the observed overactivation of inflammation is a consequence of excessive cell death post-injury, or whether excessive inflammation itself causes increased cell death at this stage and what the dynamics of this phenomenon are. Testing whether limiting apoptosis using small-molecule inhibitors affects the inflammatory response would help clarify this. ^{if} In addition see point 2

2. The authors claim that the hearts of mutants show 'unresolved fibrosis' at 21 dpci. It is unclear what the authors precisely mean by this. Typically, fibrosis refers to malignant and excessive ECM deposition by activated connective and endothelial cells, which adopt a myofibroblast identity. While increased ECM deposition is observed during regeneration, referring to regenerative ECM deposition as fibrosis does not fully consider the qualitative and quantitative differences in cell type and ECM composition between regenerative ECM and fibrotic ECM remodeling. Here, this distinction becomes crucial as the ECM type identified by transcriptional profiling to be elevated in nr4a1 mutants (e.g., Fibronectin, Collagen 12 and others) is not exclusive for fibrotic scarring but rather indicates elevated regenerative ECM expression (see Marro J et al., 2016, Bo Hu et al., 2021, Allanki S et al., 2021).

* Therefore, the authors should investigate whether nr4a1 mutants are stuck in a regenerative loop, for example because of continuously increased levels of apoptosis (see 1. S_Fig.9), leading to continued deposition of regenerative ECM, or if they are truly experiencing pathological fibrosis (increased myofibroblast levels, lysyl oxidases, collagen 1 expression). These experiments would include IHC staining for SMA, a marker for myofibroblasts and qPCR analysis of the expression levels of myofibroblast marker genes and fibrotic genes.

First revision

Author response to reviewers' comments

Response to Comments from the Reviewers:

We sincerely appreciate the valuable insights and comments provided by reviewers in this manuscript. In response, we answered all the questions and have conducted additional experiments to address nearly all of the concerns raised. We have revised the manuscript accordingly. Please see below for more details on how we addressed specific comments.

Reviewer #1:

The experiments are generally well performed and the manuscript is well written. The results are interesting, as understanding the mechanism of inflammation regulation during heart regeneration is critical for developing potential therapeutic strategies. There are several points that should be addressed in order to reinforce the conclusions proposed by the authors.

We sincerely appreciate the reviewer's positive feedback on our work. We are grateful for the recognition of our study on heart regeneration failure caused by inflammatory dysregulation. Our detailed responses to the suggestions are provided below.

Concerns:

1. The authors showed the Nr4a1 expression in macrophages and neutrophils with knock-in reporter. It will be beneficial to examine/verify whether Nr4a1 is expressed in other cell types with

published scRNA-seq datasets of zebrafish injury/regeneration heart.

We thank the reviewer for the comments. We examined *nr4a1* expression across different cell types from our previously published scRNA-seq paper (PMID 34523214). As shown in Rebuttal Figure 1, *nr4a1* is highly expressed in macrophages and neutrophils, with comparatively lower expression in erythrocytes and thrombocytes.

Rebuttal Figure 1. Expression of *nr4a1* in different immune cells from scRNA-seq data.

2. They examined PCNA/CM at 7 dpi and scar formation in regenerated heart of Nr4a1 mutant. How about the myocardial restoration in mutant at 30-60 dpi?

We appreciate the comment. To address this question, we performed MF 20 immunostaining on control and *nr4a1* mutant hearts at 30 and 60 dpci. Consistent with our scar measurement results, myocardial restoration was significantly reduced in *nr4a1* mutants compared to controls (Rebuttal Figure 2). We have now incorporated the results into the revised Fig. 2 F-I.

Rebuttal Figure 2. (A) Section images of wt or *nr4a1* mutant ventricles at 30 dpci assessed for muscle recovery, and quantification of injured area (B). (C, D) Myocardial regeneration was measured at 60 dpci in wt and *nr4a1* mutants, respectively. 10 individual samples in each group were examined. Myocardial regeneration is categorized as follows: 1=complete regeneration of a new myocardial wall; 2=partial regeneration; and 3=a strong block in regeneration. A two-tailed unpaired t-test is used in B. The Chi-squared test was performed in D.

3. The Nr4a1 is dynamically expressed during heart regeneration. Is this expression pattern correlated with inflammatory/regeneration events at different stages of heart recovery?

We thank the reviewer for pointing this out. As previously shown, *nr4a1* expression was significantly upregulated at 2 days post-cryoinjury (dpci) compared to uninjured hearts, followed by decreased expression at 7 dpci and a resurgence at 14 dpci (Fig. 1A). In response to the reviewer's suggestion, we examined additional time points and found that *nr4a1* expression declined again at 21 and 30 dpci, indicating a progressive downregulation as heart regeneration advances.

Rebuttal Figure 3. Expression pattern of *nr4a1* at different stages post-injury.

(Rebuttal Figure 3). Furthermore, immunostaining for GFP, IB4 (macrophage marker), and Mpx (neutrophil marker) in *TgKI(nr4a1-eGFP)* hearts at various time points post-injury revealed clear colocalization between GFP signals and both macrophage and neutrophil markers (shown in Rebuttal Figure 6). The above results suggest that *nr4a1* expression generally corresponds with inflammation and regeneration dynamics following injury. We have also revised our manuscript accordingly (Fig. 1A and new Fig. S1).

4. The authors stated that "significantly higher expression levels of the inflammatory-related genes *tnfa* ...". Please clarify whether there are more *tnfa*⁺ cells or enhanced *tnfa* mRNA level in *tnfa*⁺ cells.

We appreciate the suggestion. In response, we conducted RNAscope *in situ* hybridization for *tnfa*, followed by immunostaining against GFP in *TgBAC(tnfa:gfp)* transgenic hearts at 7 dpci (Rebuttal Figure 4A). As expected, an increased number of *tnfa:eGFP*⁺ cells at the injury sites were consistently observed in the *nr4a1* mutant hearts (Rebuttal Figure 4B). Subsequent analysis of *tnfa* mRNA signals, normalized to the number of *tnfa:eGFP*⁺ cells, revealed a significant increase of *tnfa* mRNA levels at the injury area in *nr4a1* mutant hearts compared to controls (Rebuttal Figure 4C). These results indicate that the number of *tnfa:eGFP*⁺ cells and *tnfa* mRNA levels are elevated in *nr4a1* mutant hearts compared to control hearts. We have added these results to the revised Fig. S5.

Rebuttal Figure 4. (A) Concurrent RNAscope *in situ* hybridization for *tnfa* and immunostaining for GFP on 7 dpci sections from wt and *nr4a1* mutant fish carrying *tnfa:eGFP* transgene. (B, C) Quantification of the number of *tnfa:eGFP*⁺ cells and the relative *tnfa* mRNA signals normalized to *tnfa:eGFP*⁺ cells, respectively. A two-tailed unpaired t-test is used. P-values<0.05 were considered statistically significant. Scale bar=75 μm.

5. What's the effects of *tnfa*^{+/-} rescue on neutrophil and macrophage responses in *Nr4a1*^{-/-} regenerating hearts?

We appreciate the reviewer's helpful comment. In response, we performed immunostaining for Mpx and IB4 to label neutrophils and macrophages in *nr4a1*^{-/-}*tnfa*^{+/-} hearts, respectively. As shown in Rebuttal Figure 5, the hearts of the *nr4a1*^{-/-}*tnfa*^{+/-} compound mutants had a significantly reduced number of neutrophils and macrophages compared to *nr4a1* mutant at 7 dpci and 14 dpci, which is consistent with the decreased inflammatory level in the compound mutants. These results are now included in the manuscript as the new Fig. S14.

Rebuttal Figure 5. (A, B) Representative cardiac section images of neutrophils distribution in three groups at 7 dpci and 14 dpci, respectively, stained with antibodies against Mpx; (C, D) Quantification of neutrophil numbers at 7 and 14 dpci, respectively; (E, F) isolectin-B4 (IB4) stained macrophages distribution at 7 dpci and 14 dpci, respectively; (G, H) Quantification of macrophage numbers at 7 and 14 dpci. Dash lines mark the injured area. Box regions show the approximate positions for quantification. two-tailed unpaired t-test are used. P-values<0.05 were considered to be statistically significant. Scale bar =100 μm.

Reviewer #2:

Most of the central findings are supported by the data and the data is generally of sufficient quality for these conclusions to be made. I support publication of the manuscript with the following suggestions:

We sincerely appreciate the reviewer's positive comments. Our responses to the suggestions are provided below.

1. Figure 1: Uninjured control sections are required of the *nr4a1KI* line. Expression appears to rise after 7dpci, later RT-PCR time points are required to determine if/when this returns to uninjured levels. A similar time course of sections comparable to that of *mpx* and *IB4* (Fig S1 and S2) would be helpful for understanding the dynamics of *nr4a1* relative to these two cell types of interest.

We thank the reviewer for the constructive suggestion. We have included the expression of *nr4a1* at 21 dpci and 30 dpci, revealing a decline in its expression as heart regeneration (Rebuttal Figure 3). To investigate the dynamics of Nr4a1 expression during cardiac regeneration, we performed immunostaining for GFP, IB4, and Mpx in *TgKI(nr4a1-eGFP)* hearts at different time points post-injury (including the uninjured controls). As shown in Rebuttal Figure 6, Nr4a1 follows a similar expression trend to macrophages (labeled with IB4) and neutrophils (Mpx-positive) expansion dynamics, which peaked at 2 Dpci and then significantly decreased after 7 dpci. We have also added these results to the new Fig. S1.

Rebuttal Figure 6. (A) Fluorescence signals from immuno-labeled Nr4a1 positive cells (GFP; green), neutrophils (Mpx; blue), and macrophages (IB4; red) in and near the wound area at multiple time points after injury, respectively. (B-D) Temporal dynamics of Nr4a1 positive cell, neutrophil, and macrophage number across different stages after injury, respectively. Dash lines mark the injured area. A two-tailed unpaired t-test is used. P-values < 0.05 were considered statistically significant. Scale bar = 100 μ m.

2. Figure 1: Quantification would be helpful to better interpret later results: relative number of neutrophils and macrophage that are positive/negative for *nr4a1+*. Triple labeling should be used to determine if *nr4a1* is only expressed by neutrophils or macrophage after injury.

We sincerely thank the reviewer for the insightful suggestion. Our triple-labeling results demonstrate that Nr4a1 is co-localized with macrophage (yellow) and neutrophil (cyan) markers, confirming its expression in both cell types (Rebuttal Figure 7A). The quantification results are presented in Rebuttal Figures 7, B, and C. Interestingly; we also observed some triple-positive signals (white), which may result from IB4-positive macrophages engulfing Mpx-positive neutrophils (PMID 31702553). In addition, some Nr4a1+ signals were detected outside of macrophages and neutrophils, consistent with our previous scRNA-seq data suggesting the

potential expression of Nr4a1 in other cell types. We have also revised our manuscript accordingly (Fig. S1).

Rebuttal Figure 7. (A) Fluorescence signals from immuno-labeled Nr4a1 positive cells (GFP; green), neutrophils (Mpx; blue), and macrophages (IB4; red) in and near the wound area at multiple time points after injury, respectively. (B, C) Temporal dynamics of double positive cell numbers across different stages after injury, respectively. Dash lines mark the injured area. A two-tailed unpaired t-test is used. P-values<0.05 were considered statistically significant. Scale bar =275 μ m.

3. Line 149/Figure 4: "A substantial accumulation of inflammatory macrophages was observed at the injury area of the control hearts at 7 dpci." This statement is not well supported without the use of definitive macrophage marker co-labeling with *tnfa*. Other cells, including activated neutrophils, have been reported to express *tnfa*.

We thank the reviewer for the suggestion. In response, we performed co-immunostaining with IB4 to label macrophages in *TgBAC(tnfa:gfp)* transgenic hearts and observed that the majority of the *tnfa*⁺ cells were co-localized with IB4⁺ macrophages, especially in the injury area (Rebuttal Figure 8A). These findings indicate a substantial accumulation of inflammatory macrophages in the post-injured hearts.

4. Line 327: "Therefore, the sustained activation of inflammatory macrophages in *nr4a1* mutants may contribute to the persistence of activated fibroblasts in the wounded area of the heart, potentially exacerbating fibrosis and impairing tissue repair." Again, there is a lack of data showing that activated/*tnfa*⁺ cells are macrophage in support of this statement. Further characterization of the *tnfa*⁺ cells is required.

We appreciate the comment and have quantified the number of *tnfa* :GFP⁺/IB4⁺ cells in *nr4a1*

mutants and controls at 7 dpci. Consistent with our previous results, this analysis further confirmed the increased number of proinflammatory macrophages in *nr4a1* mutants (Rebuttal Figure 8B). These results provide additional evidence for sustained inflammatory signaling in the absence of *nr4a1* and have been included in the new Fig. S4 A, B.

Rebuttal Figure 8. (A) Immunohistochemistry for GFP coupled with IB4 staining on 7 dpci sections from wt and *nr4a1* mutant fish carrying *tnfa:gfp* transgene. (B) Quantification of the number of *tnfa*⁺/IB4⁺ cells in the injured area. A two-tailed unpaired t-test is used. P-values < 0.05 were considered statistically significant. Arrows mark the *tnfa*⁺/IB4⁺ cells. Scale bar = 25 μ m.

5. Figure 7G: There is an inconsistency with the relative *mpeg1.1* gene expression level in *nr4a1* mutants between Figure 7G (not changed) and Figure 4G (significantly increased). This should be resolved.

We appreciate the reviewer for pointing this out. To address this, we increased the number of replicates in Figure 7G to more robustly assess *mpeg1.1* expression across different groups. The updated results revealed a significant increase in *mpeg1.1* expression in *nr4a1* mutants, consistent with our findings in Figure 4G and the RNA-seq data. We have updated Figure 7G accordingly in the revised manuscript.

6. Line 278/Figure 7I: *col12a1a* is included in the analysis but not mentioned in the text (also in Figure S7). This is not a recognized active fibroblast marker and may be pro-regenerative. Comments on the significance and interpretation of these results should be included.

We thank the reviewer for the thoughtful comment. We agree that *col12a1a* is not a canonical marker of active fibroblasts, and it has been implicated in regenerative processes. Specifically, *col12a1a* is transiently upregulated during zebrafish heart regeneration and contributes to extracellular matrix organization as a matrix-bridging component (Marro et al., 2016; Izu et al., 2023). Hu et al. (2022) also identified *col12a1a*-expressing cells in the regenerative niche. In our dataset, *col12a1a* is significantly upregulated in *nr4a1* mutant hearts, particularly at later stages post-injury (Figure 7I, Figure S7), alongside other pro-regenerative ECM genes such as *fn1a* and *postnb*. This suggests an aberrant activation of a regenerative ECM program in the mutants. However, this is accompanied by a concurrent and sustained upregulation of profibrotic genes (e.g., *egr1*, *elnb*, *egr2b*, *loxa*) at 21 and 30 dpci (new Figure 6B, Rebuttal Figure 9), indicating that the ECM remodeling is dysregulated and skewed toward pathological fibrosis. We have added text to clarify the significance of *col12a1a* and its expression pattern in the manuscript (Lines 313, 357-362).

Rebuttal Figure 9. qPCR of profibrotic ECM genes in the hearts of indicated genotypes. P-values < 0.05 were considered to be statistically significant.

Minor concerns

7. Line 63/64: Reference(s) required.

The reference has been included in the revised manuscript.

8. Line 186: "Collectively, our data indicated that *nr4a1* mutant hearts exhibit impaired resolution of fibrosis, which is associated with compromised myocardial regeneration." It should be more clearly worded to better reflect that the associated data suggest that there is increased fibrotic accumulation rather than defective or slower breakdown of fibrosis.

We thank the reviewer for the helpful suggestion. To more accurately reflect the data, we have revised the sentence to: "Collectively, our data indicated that *nr4a1* mutant hearts exhibit increased fibrotic accumulation, which is associated with compromised myocardial regeneration." (Lines 209-210).

9. Line 235: *cxcl12a* or *b* should be specified, only *cxcl12a* is known to signal to neutrophils in zebrafish.

We thank the reviewer for the clarification. We have revised the text to specifically refer to *cxcl12a*.

10. Line 430-435 and 452-458 etc: There are multiple antibodies for these proteins/antigens available from these companies. Catalogue numbers or other unique identifiers are required to indicate which was used in this study.

We thank the reviewer for the comment. As suggested, we have added the catalog numbers and relevant details for all antibodies used in this study (Lines 471-477; Lines 498-506).

Reviewer #3:

The work is of high quality and interesting to a broad range of scientists working in the field. Overall, it aligns well with previous findings on *Nr4a1* in other model organisms and contributes to

the identification of Nr4a1 function in regenerative biology. I did not identify major critical problems with the presented results or the general interpretation. However, some aspects appear unclear and will need further experimental investigations, particularly those regarding fibrosis and apoptosis.

We sincerely thank the reviewer for the positive feedback and thoughtful suggestions. We address each point in detail below.

Comments:

1. The authors show that nr4a1 mutants present strongly elevated levels of apoptotic cells 7 days post-injury (Fig S9). Since NR4A1 has been directly linked to apoptosis in other systems, including mammalian models, it remains unclear whether the observed overactivation of inflammation is a consequence of excessive cell death post-injury, or whether excessive inflammation itself causes increased cell death at this stage and what the dynamics of this phenomenon are. Testing whether limiting apoptosis using small-molecule inhibitors affects the inflammatory response would help clarify this. In addition see point 2

We thank the reviewer for the valuable suggestion. To address this question, we used pomalidomide (Pom) to suppress the inflammatory response and the apoptosis inhibitor from Sigma (Cat#178488) to block cell death. Treatment with Pom reduced the number of apoptotic cells at the injury site in *nr4a1* mutants to levels comparable to wildtype controls at 7 dpci (Rebuttal Figure 10; new Figure S12A, B). In contrast, inhibiting apoptosis had no significant impact on the inflammatory response in either wildtype or *nr4a1* mutant hearts (Rebuttal Figure 10; new Figure S12C, D). These results suggest that excessive inflammation is likely responsible for the increased apoptosis in *nr4a1* mutants.

Rebuttal Figure 10. (A) TUNEL staining in wt and *nr4a1* mutants with Pom treatment at 7 dpci. (B) Quantification of TUNEL+ cells at 7 dpci. (C) IB4 staining in wt and *nr4a1* mutants after apoptosis inhibitor treatment at 7 dpci. (D) Quantification of IB4+ cells at 7 dpci. Symbols show the sample number.

A two-tailed unpaired t-test is used. P-values < 0.05 were considered statistically significant. Scale bar in A = 275 μm, C = 100 μm.

2. The authors claim that the hearts of mutants show 'unresolved fibrosis' at 21 dpci. It is unclear what the authors precisely mean by this. Typically, fibrosis refers to malignant and excessive ECM deposition by activated connective and endothelial cells, which adopt a myofibroblast identity. While increased ECM deposition is observed during regeneration, referring to regenerative ECM

deposition as fibrosis does not fully consider the qualitative and quantitative differences in cell type and ECM composition between regenerative ECM and fibrotic ECM remodeling. Here, this distinction becomes crucial as the ECM type identified by transcriptional profiling to be elevated in *nr4a1* mutants (e.g., Fibronectin, Collagen 12 and others) is not exclusive for fibrotic scarring but rather indicates elevated regenerative ECM expression (see Marro J et al., 2016, Bo Hu et al., 2021, Allanki S et al., 2021).

* Therefore, the authors should investigate whether *nr4a1* mutants are stuck in a regenerative loop, for example because of continuously increased levels of apoptosis (see 1. S_Fig.9), leading to continued deposition of regenerative ECM, or if they are truly experiencing pathological fibrosis (increased myofibroblast levels, lysyl oxidases, collagen 1 expression). These experiments would include IHC staining for SMA, a marker for myofibroblasts and qPCR analysis of the expression levels of myofibroblast marker genes and fibrotic genes.

We thank the reviewer for this insightful comment. To clarify whether *nr4a1* mutants exhibit pathological fibrosis or are instead trapped in a prolonged regenerative ECM state, we performed additional experiments as suggested. First, we conducted immunostaining for α -SMA at 30 and 60 dpci. As shown in Rebuttal Figure 11, *nr4a1* mutant hearts exhibit increased α -SMA deposition surrounding the injury area, suggesting elevated myofibroblast activation.

We also examined the expression of established profibrotic genes, including lysyl oxidases and *col1a1a*. Many of these markers were significantly upregulated in *nr4a1* mutants at both 21 and 30 dpci (Rebuttal Figure 9), consistent with activation of fibrotic pathways.

Together, these results indicate that *nr4a1* mutant hearts undergo aberrant ECM remodeling that involves both regenerative and pathological fibrotic components. This suggests a disrupted transition between regenerative and resolution phases, rather than a sustained regenerative ECM response. We have incorporated these new findings and related discussion into the revised manuscript (new Figure 5J-M; Lines 197-204, 357-362). Future work will explore the regulatory mechanisms underlying this dysregulation.

Rebuttal Figure 11. (A, C) α -SMA immunostaining at 30 dpci and 60 dpci injured hearts, respectively. Left: wt group; Right: *nr4a1* mutant group. (B, D) Quantification of α -SMA signal in the wounded area at 30 dpci and 60 dpci, respectively. Symbols show the individual sample number. A two-tailed unpaired t-test is used. P-values<0.05 were considered to be statistically significant. Scale bar=275 μ m.

Second decision letter

MS ID#: dev.204395R1

MS TITLE: Nr4a1 Modulates Inflammation and Heart Regeneration in Zebrafish

AUTHORS: Jiandong Liu; Dong Feng; Yanhan Dong; Yiran Song; Yifang Xie; Brian Spurlock; Landry Kuehn; Nicholas Yapundichl; Li Qian; Tingting Lyu

Dear Jiandong,

I have now received all the referees reports on the above manuscript, and have reached a decision. The referees' comments are appended below, or you can access them online: please go to .

The overall evaluation is positive and we would like to publish a revised manuscript in Development, provided that the referees' comments can be satisfactorily addressed. Please attend to all of the reviewers' comments in your revised manuscript and detail them in your point-by-point response. Reviewers #1 and #2 recommend acceptance of the manuscript, and Reviewer #3 has comments that I expect you could address in a final minor revision.

Reviewer 1

The authors have adequately addressed my concerns and I have no more comments.

Reviewer 3

Most of my concerns have been addressed by the authors.

However I am somewhat surprised by the used experimental design:

1. In the revised version, the authors used the small molecule inhibitor POM, a Tnf- $\hat{\pm}$ inhibitor to suppress inflammation.

The authors compared between Pom treated WT and Pom treated nr4a1^{-/-} mutants which may not be the most adequate way to demonstrate the effect of reduced inflammatory response on apoptosis. A more informative approach would have been the comparison of untreated and Pom treated nr4a1^{-/-} hearts directly. Observing a significant reduction in the number of apoptotic cells within the mutant upon Pom treatment would more convincingly indicate that the increased apoptosis is driven by an increased inflammatory response in the absence of Nr4a1.

2. When mentioning a comparison towards WT (see below), it may be more appropriate to evaluate Pom-treated nr4a1^{-/-} tissue to vehicle control WT instead of Pom treated WT. Alternatively it should be at least more clearly stated that there are no significant differences between vehicle and Pom treated WT.

"Treatment with Pom reduced the number of apoptotic cells at the injury site in nr4a1 mutants to levels comparable to wildtype controls at 7 dpci (Rebuttal Figure 10; new Figure S12A, B)."

This is further underscored by the qualitative observations in Figure 10A, where Pom-treated WT samples appear to exhibit a higher number of apoptotic cells compared to untreated WT controls— independent of the quantification presented in panel B.

However, along with the additional data showing that a reduction of apoptosis does not lead to a reduction in inflammation I do not expect the interpretation to be of substantial difference in a direct comparison. Nevertheless, the interpretation of the presented comparison remains somewhat difficult.

Off note: the label of the y-axis in the new figure panel 5 K and M (quantification of SMA) is not adequately explained. This should be corrected.

Second revision

Author response to reviewers' comments

Response to Comments from the Reviewers:

We thank all the reviewers for their time and effort in evaluating our revised manuscript and sincerely appreciate the additional valuable comments provided by reviewer #3. In response, we have conducted further experiments to address the concerns raised and revised the manuscript accordingly.

Please see below for more details on how we addressed specific comments.

Reviewer #3: Most of my concerns have been addressed by the authors. However I am somewhat surprised by the used experimental design:

1. In the revised version, the authors used the small molecule inhibitor POM, a Tnf- α inhibitor to suppress inflammation. The authors compared between Pom treated WT and Pom treated $nr4a1^{-/-}$ mutants which may not be the most adequate way to demonstrate the effect of reduced inflammatory response on apoptosis. A more informative approach would have been the comparison of untreated and Pom treated $nr4a1^{-/-}$ hearts directly. Observing a significant reduction in the number of apoptotic cells within the mutant upon Pom treatment would more convincingly indicate that the increased apoptosis is driven by an increased inflammatory response in the absence of Nr4a1.

Rebuttal Figure 1. (A) TUNEL staining in wt and $nr4a1$ mutants with Pom treatment at 7 dpci. (B) Quantification of TUNEL+ cells at 7 dpci. A two-tailed unpaired t-test is used. P-values < 0.05 were considered statistically significant. Scale bar = 275 μ m.

We thank the reviewer for the insightful comment. To further strengthen our conclusions, we performed an additional experiment focusing on directly comparing untreated and Pom-treated $nr4a1^{-/-}$. The results showed a significant reduction in the number of apoptotic cells in the $nr4a1^{-/-}$ hearts following Pom treatment compared to untreated mutants (Rebuttal Figure 1; new Figure S12). As the reviewer suggested, the finding provides further evidence that the elevated apoptosis observed in $nr4a1$ mutants is driven by an exacerbated inflammatory response.

2. When mentioning a comparison towards WT (see below), it may be more appropriate to evaluate Pom-treated $nr4a1^{-/-}$ tissue to vehicle control WTs instead of Pom treated WT. Alternatively it should be at least more clearly stated there are no significant differences between vehicle and Pom treated WTs. "Treatment with Pom reduced the number of apoptotic cells at the injury site in $nr4a1$ mutants to levels comparable to wildtype controls at 7 dpci (Rebuttal Figure 10; new Figure S12A, B)." This is further underscored by the qualitative observations in Figure 10A, where Pom-treated WT samples appear to exhibit a higher number of apoptotic cells compared to untreated WT controls— independent of the quantification presented in panel B. However, along with the additional data showing that a reduction of apoptosis does not lead to a reduction in inflammation I do not expect the interpretation to be of substantial difference in a direct comparison. Nevertheless, the interpretation of the presented

comparison remains somewhat difficult.

We thank the reviewer for this comment and conducted additional comparative analysis to address it. As shown in Rebuttal Figure 1, there was no significant difference in apoptotic cell numbers between Pom-treated *nr4a1*^{-/-} hearts and vehicle-treated wild-type controls or between Pom-treated and vehicle-treated wild-type hearts. These findings further suggest that inhibiting inflammation can partially rescue the elevated apoptosis associated with *nr4a1* deficiency. We have updated the representative images in Fig. S12A to more accurately reflect the quantification data.

Similarly, we expanded the comparative analysis to include multiple groups treated with apoptosis inhibitor. As shown in Rebuttal Figure 2, inhibiting apoptosis had no significant effect on the inflammatory response in either wild-type or *nr4a1* mutant hearts. These results suggest that excessive inflammation is more likely responsible for the increased apoptosis in *nr4a1*^{-/-} mutants.

Rebuttal Figure 2. Quantification of IB4+ cells at 7 dpci. Symbols show the sample number. A two-tailed unpaired t-test is used. P-values < 0.05 were considered statistically significant.

Off note: the label of the y-axis in the new figure panel 5 K and M (quantification of SMA) is not adequately explained. This should be corrected.

We thank the reviewer for pointing it out. We have corrected it.

Third decision letter

MS ID#: dev.204395R2

MS TITLE: Nr4a1 Modulates Inflammation and Heart Regeneration in Zebrafish

AUTHORS: Jiandong Liu; Dong Feng; Yanhan Dong; Yiran Song; Yifang Xie; Brian Spurlock; Landry Kuehn; Nicholas Yapundichl; Li Qian; Tingting Lyu

ARTICLE TYPE: Research Article

Dear Jiandong,

I am happy to tell you that your manuscript has been accepted for publication in Development, pending our standard publication integrity checks.